# Leveraging AutoML for Sustainable Deep Learning: A Multi-Objective HPO Approach on Deep Shift Neural Networks

**Leona Hennig**    *l.hennig@ai.uni-hannover.de*
*Institute of Artificial Intelligence*
*Leibniz University Hanover*

**Marius Lindauer**    *m.lindauer@ai.uni.hannover.de*
*Institute of Artificial Intelligence*
*Leibniz University Hanover*
*L3S Research Center*

**Reviewed on OpenReview:** *https://openreview.net/forum?id=vk7b11DHcW*

## Abstract

Deep Learning (DL) has advanced various fields by extracting complex patterns from large datasets. However, the computational demands of DL models pose environmental and resource challenges. Deep Shift Neural Networks (DSNNs) present a solution by leveraging shift operations to reduce computational complexity at inference. Compared to common DNNs, DSNNs are still less well understood and less well optimized. By leveraging AutoML techniques, we provide valuable insights into the potential of DSNNs and how to design them in a better way. We focus on image classification, a core task in computer vision, especially in low-resource environments. Since we consider complementary objectives such as accuracy and energy consumption, we combine state-of-the-art multi-fidelity (MF) hyperparameter optimization (HPO) with multi-objective optimization to find a set of Pareto optimal trade-offs on how to design DSNNs. Our approach led to significantly better configurations of DSNNs regarding loss and emissions compared to default DSNNs. This includes simultaneously increasing performance by about 20% and reducing emissions, in some cases by more than 60%. Investigating the behavior of quantized networks in terms of both emissions and accuracy, our experiments reveal surprising model-specific trade-offs, yielding the greatest energy savings. For example, in contrast to common expectations, quantizing smaller portions of the network with low precision can be optimal with respect to energy consumption while retaining or improving performance. We corroborated these findings across multiple backbone architectures, highlighting important nuances in quantization strategies and offering an automated approach to balancing energy efficiency and model performance.

## 1 Introduction

Deep Learning (DL) is a promising approach to extracting information from large datasets with complex structures. This includes performing computations in IoT environments and on edge devices (Li et al., 2018; Zhou et al., 2019), which can come with strict limitations on energy consumption. This is especially true for the ever-increasing size and performance of such models due to the progress in science and industry. Running these models is not free of computational costs (Sze et al., 2017), and minimizing this cost directly affects the environmental impact of a model (Schwartz et al., 2020). Even if resource consumption should not be considered crucial in view of environmental impact, efficiently designed neural networks free up resources that can be used for other tasks, e.g., edge computing or computations for advanced driver assistance systems (Howard et al., 2017). With our approach, we contribute to DL by minimizing its environmental footprint and allowing applications in low-resource settings.

In this context, we focus on image classification, one of the core tasks in computer vision and particularly relevant for low-power applications. Image classification is widely common in scenarios where resources are limited, such as edge computing, automated driving, and industrial production environments. These applications require fast, accurate, and energy-efficient models that can process visual information in real time. Because it is well understood and broadly benchmarked, image classification enables reproducible evaluation of architectural and optimization strategies with minimal confounding of task-specific variables.

Of particular interest to us are Deep Shift Neural Networks (DSNNs), which offer great potential to reduce power consumption compared to traditional DL models, e.g., by reducing the inference time by a factor of 4 (Elhoushi et al., 2021). Instead of expensive floating-point arithmetic, they leverage cheap shift operations - specifically, bit shifting - as the computational unit, which boosts efficiency by replacing costly multiplication operations in convolutional networks. Although DSNNs offer great promise, so far, different design decisions, including training hyperparameters and shift architecture, have not been well studied, and there is little knowledge about their full potential. We suspect that the configuration of DSNNs has a huge impact on both performance and computational efficiency.

One of the key challenges with DSNNs is determining the appropriate level of precision for shift operations to minimize quantization errors without excessively increasing the computational load. To address this challenge, we propose to apply automated machine learning (AutoML) to DSNNs to find their optimal configuration. This is achieved by hyperparameter optimization (HPO) (Bischl et al., 2023) and a neural architecture search on a macro-level (Elsken et al., 2019). Integrating multi-fidelity (MF) and multi-objective optimization (MO) techniques (Belakaria et al., 2020) facilitates an optimal exploration of the configuration space that trades off predictive performance and energy consumption (Deb, 2014). To this end, we extended the SMAC3 approach (Lindauer et al., 2022), a state-of-the-art HPO package (Eggensperger et al., 2021), so that its MO implementation effectively balances the trade-off between achieving high predictive accuracy and minimizing energy consumption. Using tools such as CodeCarbon (Lacoste et al., 2019; Lottick et al., 2019) during the training and evaluation phases provides insight into the energy consumption and carbon emissions associated with each model configuration. The MF aspect allows for the efficient use of computational resources by evaluating configurations at varying levels of approximation. Our work is in the spirit of Green AutoML (Tornede et al., 2023), considering efficient AutoML to gain insight into the design of efficient DSNNs.

**Contributions** Overall, we contribute to Green AutoML w.r.t. DSNNs by:

1. Specifying the first configuration space tailored to DSNNs, which is efficiently optimized by a combination of multi-objective and multi-fidelity AutoML approaches;

2. Empirically exploring how specific design choices in DSNNs lead to different trade-offs between accuracy and energy efficiency, enabling stakeholders and researchers to leverage these findings to develop energy-efficient applications that maintain high computational accuracy; and

3. Identifying specific configurations of DSNNs that surpass the baseline results in both dimensions of the performance-efficiency optimization problem.

The corresponding repository is available at `https://github.com/automl/Auto-DSNN`.

## 2 Related Work

Both multi-fidelity optimization (MF) (Bischl et al., 2023) and multi-objective optimization (MO) (Morales-Hernández et al., 2022) for AutoML have gotten a lot of traction in recent years. The combined integration of multi-fidelity multi-objective optimization (MFMO) has seen some advancements to enhance the efficiency of model training while minimizing environmental impact. Belakaria et al. (2020) proposed an acquisition function based on output space entropy search for multi-fidelity multi-objective Bayesian optimization (MFMO-BO-OSES). Their method addresses the exploration-exploitation dilemma by prioritizing the acquisition of data points that significantly reduce the entropy of the Pareto front. This approach enables more strategic sampling decisions and leverages lower-fidelity evaluations to approximate the Pareto front effectively, aligning

with the sustainability goals of Green AutoML. Similarly, Schmucker et al. (2020) considered a combination of multi-objective and multi-fidelity optimization but focused on fairness as the second objective. With our MFMO approach, we contribute an algorithm tailored specifically for performing efficient HPO tasks using BO, directly minimizing emissions in the process.

A well-explored technique for reducing the computational complexity of neural networks themselves is network quantization. It involves lowering the precision of weights and activations, which decreases the model's memory footprint and accelerates inference. Works by Courbariaux et al. (2015) and Rastegari et al. (2016), for example, have demonstrated that techniques such as BinaryConnect and XNOR-Net not only reduce computational requirements but also maintain near-state-of-the-art performance, underscoring the potential of quantization to balance performance with computational efficiency. This has also been transferred into the field of LLMs, where 1-Bit transformer architectures are used to address the challenges of increasing model size (Wang et al., 2023). Strongly related to network quantization, Deep Shift Neural Networks (DSNNs), proposed by Elhoushi et al. (2021), represent an advancement towards quantization combined with efficient operators. DSNNs employ bitwise shift operations instead of traditional multiplications, thus further reducing the computational overhead and power consumption. This innovation is particularly crucial for deploying Deep Learning models in power-sensitive or resource-constrained environments, further contributing to the environmental sustainability of AI technologies.

DSNNs offer a more balanced trade-off between computational efficiency and accuracy compared to binary methods like BinaryConnect and XNOR-Net by replacing multiplications with bitwise shifts, which are more power-efficient yet maintain greater precision, reducing the accuracy loss typically associated with binary quantization. This allows DSNNs to achieve better performance in resource-constrained environments while still minimizing computational overhead (Elhoushi et al., 2021).

AdderNet (Chen et al., 2020a) is another approach for reducing the amount of computationally expensive multiplications during network training and inference by using the $\ell_1$-norm distance between input and filter vectors to compute activations. There are efforts in mixed-precision quantization, where different bit-widths are assigned to different layers or channels of the network. It allows for higher precision where necessary and lower precision where possible, optimizing the trade-off between accuracy and efficiency (Motetti et al., 2024). Similarly, ternary quantization (TTQ) constrains the weights to three discrete values: $\{-1, 0, +1\}$. To mitigate the accuracy loss from reduced precision, TTQ introduces learned scaling factors, allowing networks to maintain performance comparable to their full-precision counterparts while significantly reducing memory and power usage (Rokh et al., 2023). HPO combined with DSNNs further optimizes both performance and resource efficiency by tailoring shift depths and quantization strategies, allowing for fine-tuned control over energy consumption and accuracy in constrained environments, making them ideal for mixed-precision and quantization-aware neural network applications.

## 3 Background

The following chapter introduces foundational concepts for our approach.

### 3.1 Deep Shift Neural Networks

A Deep Shift Neural Network (DSNN) is an approach to reduce the computational and energy demands of Deep Learning (Elhoushi et al., 2021). DSNNs achieve a considerable reduction in latency time by simplifying the network architecture such that they replace the traditional multiplication operations in neural networks with bit-wise shift operations and sign flipping, making DSNNs suitable for computing devices with limited resources. There are two methods for training DSNNs (Elhoushi et al., 2021): DeepShift-Q (Quantization) and DeepShift-PS (Powers of two and Sign). DeepShift-Q involves training regular weights constrained to powers of 2 by quantizing weights to their nearest power of two during both forward and backward passes. In DeepShift-Q, the weights are quantized to powers of two by rounding the logarithm of the absolute weights to the nearest integer. This process simplifies the weight representation and ensures compatibility with bitwise shift operations. The sign is then applied to preserve the original weight polarity. DeepShift-PS directly includes the values of the shifts and sign flips as trainable hyperparameters, offering finer control over weight

adaptation. This approach removes the need for explicit rounding during training, potentially leading to improved precision at the cost of additional parameter updates.

The DeepShift-Q approach obtains the sign matrix $S$ from the trained weight matrix $W$ as $S = sign(W)$. The power matrix $P$ is the base-2 logarithm of $W$'s absolute values, i.e., $P = \log_2(|W|)$. After rounding $P$ to the nearest power of two, $\tilde{P} = round(P)$, the quantized weights $\tilde{W}_q$ are calculated by applying the sign $S$:

$$\tilde{W}_q = flip(2^{\tilde{P}}, S) \,. \tag{3.1}$$

The DeepShift-PS approach optimizes neural network weights by directly adapting the shift ($\tilde{P}$) and sign ($\tilde{S}$) values. The shift matrix $\tilde{P}$ is obtained by rounding the base-2 logarithm of the weight values, $\tilde{P} = round(P)$, and the sign flip $\tilde{S}$ is computed as $\tilde{S} = sign(round(S))$. Weights are calculated as

$$\tilde{W}_{ps} = flip(2^{\tilde{P}}, \tilde{S}) \,, \tag{3.2}$$

where the sign flip operation $\tilde{S}$ assigns values of $-1$, $0$, or $+1$ based on $S$. Directly training shift and sign values could allow for more precise control in optimizing the network's computational efficiency by reducing mathematical imprecision. However, training the floating point weights and only rounding them during the forward and backward passes might increase the precision and reduce the error in training the weights.

## 3.2 Hyperparameter Optimization

The increasing complexity of Deep Learning algorithms enhances the need for automated hyperparameter optimization (HPO) to increase model performance (Bischl et al., 2023). Consider a dataset $\mathcal{D} = \{(x_i, y_i)\}_{i=1}^{N} \in \mathbb{D} \subset \mathcal{X} \times \mathcal{Y}$, where $\mathcal{X}$ is the instance space and $\mathcal{Y}$ is the target space, and a hyperparameter configuration space $\Lambda = \{\lambda_1, \dots, \lambda_L\}$, $L \in \mathbb{N}$. In our work, $\mathcal{M}$ denotes the space of possible DSNN models. The dataset $\mathcal{D}$ is split into disjoint training, validation, and test sets: $\mathcal{D}_{train}, \mathcal{D}_{val}$, and $\mathcal{D}_{test}$ respectively. An algorithm $\mathcal{A} : \mathbb{D} \times \Lambda \to \mathcal{M}$ trains a model $M \in \mathcal{M}$, instantiated with a configuration of $L$ hyperparameters sampled from $\Lambda$, on the training data $\mathcal{D}_{train}$. The performance of the algorithm is assessed via an expensive-to-evaluate loss function $\mathcal{L} : \mathcal{M}_\lambda \times \mathbb{D} \to \mathbb{R}$, which involves both the training on $\mathcal{D}_{train}$ and the evaluation of the model on $\mathcal{D}_{val}$. The optimization objective of HPO is to find configurations $\lambda^* \in \Lambda$ with minimal validation loss $\mathcal{L}$:

$$\lambda^* \in \arg\min_{\lambda \in \Lambda} \mathcal{L}\big(\mathcal{A}(\mathcal{D}_{train}, \lambda), \mathcal{D}_{val}\big). \tag{3.3}$$

Finally, the model's final performance is assessed on $\mathcal{D}_{test}$.

## 3.3 Bayesian Optimization

For a given dataset, Bayesian Optimization (BO) for HPO is a strategy for global optimization of black-box loss functions $\mathcal{L} : \mathcal{M}_\lambda \times \mathbb{D} \to \mathbb{R}$ that are expensive to evaluate (Jones et al., 1998).

BO uses a probabilistic surrogate model $\mathcal{S}$ to approximate the loss function, commonly given by a Gaussian Process or a Random Forest (Rasmussen & Williams, 2006; Hutter et al., 2011; Shahriari et al., 2016). An acquisition function $\alpha : \Lambda \to \mathbb{R}$ guides the search for the next optimal evaluation points by balancing the exploration-exploitation trade-off, based on the set of previously evaluated configurations $\{(\lambda_1, \mathcal{L}(M_{\lambda_1}, \cdot)), ..., (\lambda_{m-1}, \mathcal{L}(M_{\lambda_{m-1}}, \cdot))\}$ at time $m$. Common choices for acquisition functions include expected improvement (EI) (Jones et al., 1998) since it calculates the expected improvement in the objective function value and guides the search towards regions where improvements are most likely.

Entropy-based methods like Entropy Search (ES) (Hennig & Schuler, 2012) and Predictive Entropy Search (PES) (Hernández-Lobato et al., 2014) aim to reduce the entropy of the posterior distribution of the maximizer, focusing on information-rich regions. The Knowledge Gradient (KG) (Frazier et al., 2009) offers a strategy for maximizing the expected improvement of the objective, considering all potential outcomes, valuable in scenarios with noisy measurements. BO is particularly well-suited for hyperparameter optimization in Deep Learning, where evaluating the performance of a model configuration can be computationally expensive because of the training of each configuration. BO is sample-efficient in evaluating $\mathcal{L}$ on only a few configurations.

### 3.4 Multi-Fidelity Optimization

To reduce the cost of fully training multiple DSNN configurations, we employ a multi-fidelity (MF) approach (Li et al., 2017), which is a common strategy in AutoML to navigate the trade-off between performance and approximation error (Hutter et al., 2019). MF approaches train cheap-to-evaluate proxies of black-box functions following different heuristics, e.g., allocating a small number of epochs to many configurations in the beginning and training the best-performing ones on an increasing number of epochs. Formally, we define a space of fidelities $\mathcal{F}$ and aim to minimize a function $F \in \mathcal{F}$ (Kandasamy et al., 2019):

$$\min_{\lambda \in \Lambda} F(\lambda). \tag{3.4}$$

We approximate $F \in \mathcal{F}$, using a series of lower-fidelity, i.e., less expensive approximations $\{f(\lambda)_1, \ldots, f(\lambda)_j = F(\lambda)\}$, where $j$ denotes the total number of fidelity levels. The target function $F \in \mathcal{F}$ corresponds to the loss function $\mathcal{L}$ of HPO and BO. The allocated resources for evaluating a model's performance at various fidelities are referred to as a budget, e.g., training a DNN for only $n \in \mathbb{N}$ epochs instead of until convergence. MF typically assumes that the highest fidelities approximate the black-box function best. The longer a model is trained, the more accurate its approximation of an underlying function gets.

### 3.5 Multi-Objective Optimization

Multi-objective optimization (MO) addresses problems involving multiple, often competing, objectives. This approach is used in scenarios where trade-offs between two or more conflicting objectives must be navigated, such as in the context of DSNNs, enhancing accuracy alongside reducing energy consumption. MO aims to identify Pareto optimal solutions (Deb, 2014). New points are added based on the current observation dataset $\mathcal{D}_{obs} = \{(\lambda_1, \mathcal{L}(M_{\lambda_1}, \cdot)), \ldots, (\lambda_n, \mathcal{L}(M_{\lambda_m}, \cdot)\}$ at time $m + 1$. These points augment the surface formed by the non-dominated solution set $D_n^\star$, which satisfies the condition for $d$ objective variables and a loss function $\mathcal{L} = (\mathcal{L}_1, \ldots, \mathcal{L}_d)$, where $\mathcal{L}_k$ corresponds to the loss regarding objective $k$ (Wada & Hino, 2019):

$$\forall \lambda, (\lambda, \mathcal{L}(\lambda)) \in \mathcal{D}_n^\star \subset \mathcal{D}_n, \ (\lambda', \mathcal{L}(\lambda')) \in \mathcal{D}_n \exists k \in \{1, \ldots, d\} : \ \mathcal{L}_k(\lambda) \leq \mathcal{L}_k(\lambda'). \tag{3.5}$$

W.l.o.g., we assume the minimization of all objectives. The observation dataset $\mathcal{D}_{obs}$ is iteratively updated to search for solutions that approximate the Pareto front.

## 4 Approach

Our goal is to provide deeper insights into the structure of Deep Shift Neural Networks (DSNNs). We optimize these networks with respect to both performance and efficiency, highlighting how their hyperparameters influence the optimization process and resulting trade-offs.

### 4.1 Configuration Space Exploration

The foundation of our approach lies in defining and exploring a robust configuration space tailored specifically to DSNNs. This space includes a range of hyperparameters that influence the network's performance and energy efficiency. Key hyperparameters under consideration include:

1. Shift Depth: determines the number of network layers converted to employ shift operations, replacing conventional floating point operations and thereby reducing computational overhead.

2. Shift Type: selects the method of shift operation, either quantization (DeepShift-Q) or direct training of shifts (DeepShift-PS), impacting the network's training dynamics and inference efficiency.

3. Bit Precision for Weights and Activations: influences the network's accuracy and the granularity of its computations, affecting both performance and power consumption.

4. Rounding Type: affects how weight adjustments are handled during training, with options for deterministic or stochastic rounding, each offering trade-offs in terms of stability and performance.

Table 1 details the configuration space for a ResNet20 model adapted for DSNNs, outlining the range and default values of each hyperparameter considered in our study.

## 4.2 Multi-Fidelity Multi-Objective Optimization Framework

To computationally enhance DSNNs via AutoML, we employ multi-fidelity optimization (MF), see Section 3. A well-known MF algorithm is successive halving (Jamieson & Talwalkar, 2016), where $n_c$ configurations are trained on an initial small budget $b_I$. It addresses the trade-off between $b_I$ and $n_c$, or between approximation error and exploration inherent in successive halving, using the HyperBand algorithm for MF. HyperBand (Li et al., 2017) runs successive halving in multiple brackets, where each bracket provides a combination of $n_c$ and a fraction of the total budget per configuration so that they sum up to the total budget. We extend this to multi-fidelity multi-objective optimization (MFMO). We simultaneously address the accuracy of the model as well as its energy consumption using a two-dimensional objective function:

$$\mathcal{L}_{\text{MO}} : \Lambda \to \mathbb{R}^2, \quad \mathcal{L}_{\text{MO}}(\lambda) = \big(\mathcal{L}_{\text{loss}}(\lambda), \mathcal{L}_{\text{emission}}(\lambda)\big), \tag{4.1}$$

where, given a configuration $\lambda \in \Lambda$, $\mathcal{L}_{\text{loss}}(\lambda)$ aims to minimize the loss, enhancing the model's accuracy, and $\mathcal{L}_{\text{emission}}(\lambda)$ seeks to minimize the energy consumption during training and inference, promoting environmental sustainability. We aim to solve the following optimization problem:

$$\arg\min_{\lambda \in \Lambda} \mathcal{L}_{\text{MO}}(\lambda). \tag{4.2}$$

Table 1: Configuration search space of ResNet20. The first half includes commonly used training hyperparameters for deep learning, whereas the second half is specific to DSNNs.

| Hyperparameter | Search Space | Default |
|---|:---:|:---:|
| Standard Training Hyperparameters | | |
| Batch Size | [32, 128] | 128 |
| Optimizer | {SGD, Adam, Adagrad, Adadelta, RMSProp, RAdam, Ranger} | SGD |
| Learning Rate | [0.001, 0.1] | 0.1 |
| Momentum | [0.0, 0.9] | 0.9 |
| Weight Decay | [1e-6, 1e-2] | 0.0001 |
| DSNN-Specific Hyperparameters | | |
| Weight Bits | [2, 8] | 5 |
| Activation Integer Bits | [2, 32] | 16 |
| Activation Fraction Bits | [2, 32] | 16 |
| Shift Depth | [0, 20] | 20 |
| Shift Type | {Q, PS} | PS |
| Rounding | {deterministic, stochastic} | deterministic |

Our approach leverages the MFMO framework to efficiently explore the configuration space using computationally inexpensive proxies for full training. This enables broader and more tractable exploration of the hyperparameter landscape within practical resource constraints. The method is inherently architecture- and task-agnostic, requiring no modifications when applied to different models or datasets, aside from adjusting the DSNN's quantization mechanism to align with the specific target architecture and task.

## 4.3 Algorithmic Implementation of MFMO

We use the ParEGO algorithm (Knowles, 2006) to compute Pareto optimal configurations. It transforms the multi-objective problem into a series of single-objective problems by introducing varying weights for

the objectives in each iteration of HyperBand, thus optimizing a different scalarization per evaluation to approximate the Pareto front. The resulting single-objective optimization function can then be evaluated in an MF setting. All configurations having survived a successive halving bracket are checked against the current Pareto front approximation and the Pareto set is updated if necessary. The computational strategy initially involves computing a broad array of configurations and leveraging the successive halving method to efficiently narrow down the field to the most promising candidates. We specifically target solutions that represent both extremes of the Pareto front, those that excel in one objective at the potential expense of the other, and configurations that provide a balanced compromise between the two objectives. We included pseudocode of our algorithmic implementation in Algorithm 1 in the appendix.

## 5  Experiments

In the following section, we detail the setup and methodology used to evaluate our approach discussed in Section 4, focusing on optimizing Deep Shift Neural Networks (DSNNs) through multi-fidelity, multi-objective optimization (MFMO), and extending the DSNN objective function to multi-objective to compute a Pareto front for optimality regarding performance and efficiency. We discuss how our approach successfully navigates the model performance and environmental impact trade-offs. From this, we gain insights into DSNNs and how specific design choices might affect their performance. By identifying optimal configurations for both or either objectives, we draw conclusions about how the DSNN-specific hyperparameters in the network architecture interact with each other.

### 5.1  Evaluation Setup

We train and evaluate our models on the CIFAR10 dataset (Krizhevsky et al., 2009) and the Caltech101 dataset (Fei-Fei et al., 2004). Our experiments are conducted on the NVIDIA A100, a widely used GPU, standard in research and industry for some time. Its computational capabilities and availability in many high-performance clusters make it a popular choice in the machine learning and deep learning communities. Using the A100, we ensure that our findings are broadly applicable and relevant to real-world scenarios, aligned with hardware commonly utilized for training and deploying advanced neural networks.

For hyperparameter optimization (HPO) with multi-fidelity optimization, we extend SMAC3 (Hutter et al., 2011; Lindauer et al., 2022), as well-known state-of-the-art HPO package (Eggensperger et al., 2021). As a starting point, we chose the well-known ResNet20 (He et al., 2016) architecture as used by Elhoushi et al. (2021). Overall, this architecture is well understood and allows us to study DSNNs with few confounding factors. Additionally, we evaluate our approach using the well-known GoogLeNet (Szegedy et al., 2015) and MobileNetV2 architectures (Sandler et al., 2018). We follow the model implementation of Elhoushi et al. (2021) to ensure comparability. The configuration space is given in Table 1, for which we focus on the DSNN-specific hyperparameters (lower part of the table) and general training hyperparameters (upper part). The fidelities are the number of epochs.

For multi-objective optimization, we aim to compute a Pareto front of optimal configurations for performance and energy consumption. To incorporate the environmental impact into our HPO workflow, we use the CodeCarbon emissions tracker (Lacoste et al., 2019; Lottick et al., 2019) to track carbon emissions from computational processes, in our case inference, by monitoring energy use and regional energy mix in kgCo2eq, grams of $CO_2$ equivalents. We use power consumption as a proxy for $CO_2$ emissions, acknowledging that external factors such as energy source and grid transmission losses are not directly controlled. These emission values are incorporated into SMAC3 alongside DSNN's performance metric. The loss is $1-$ accuracy.

### 5.2  Results

#### 5.2.1  Quantitative Results

We first discuss the quantitive results, then the importance of the optimized hyperparameters, and finally, the implications for the configuration space. Note that we focus on the insights gained regarding DSNNs and not on how efficient our (or others') HPO approach is. In Figure 2, we present the computed Pareto fronts of

a ResNet20, MobileNetV2 and GoogLeNet architecture, optimized with our multi-fidelity multi-objective (MFMO) framework, on the CIFAR10 and Caltech101 datasets. Shown is a diverse set of optimal configurations that either minimize or balance the primary objectives of model accuracy and energy consumption. These are aggregated results over three seeds. The Pareto front shown results from aggregating the three individual Pareto fronts and extracting the Pareto optimal points. The default value is the mean over the loss and emissions of the default configuration on the three seeds. The Pareto fronts in Figure 2 depict how each configuration performs relative to the others within the defined hyperparameter space. The configurations were evaluated regarding classification loss and emissions emitted during inference of the model instantiated with the respective configuration. Throughout this paper, when referring to emissions, we measured emissions at inference. Although we expect that Elhoushi et al. (2021) optimized their hyperparameters at least manually, we can show that our AutoML approach found even better trade-offs of the two objectives. The default configuration for the DSNN, as designed by Elhoushi et al. (2021), is in fact not part of the Pareto front in Figure 2. This holds for all architectures on both datasets. This means that there are better configurations that dominate the default configuration regarding both loss and emissions on MobileNetV2 (Figures 2a and 2b), GoogLeNet (Figures 2c and 2d) and ResNet20 (Figures 2e and 2f) on Caltech101 and CIFAR10.

Having a closer look at Figure 2f, the underlying goal of our MFMO optimization remains to balance performance and efficiency. Hence, the configurations at the bottom left of the Pareto front are especially relevant since they minimize both objectives simultaneously instead of heavily prioritizing either. There is an absolute reduction in loss of up to 20% in these configurations compared to the defaults. At the same time, relative emission reduction ranges from approx. 10% to more than 60% in Figures 2b and 2e. Additionally, we achieved a maximum loss reduction of about 20% for the ResNet20 on CIFAR10. This

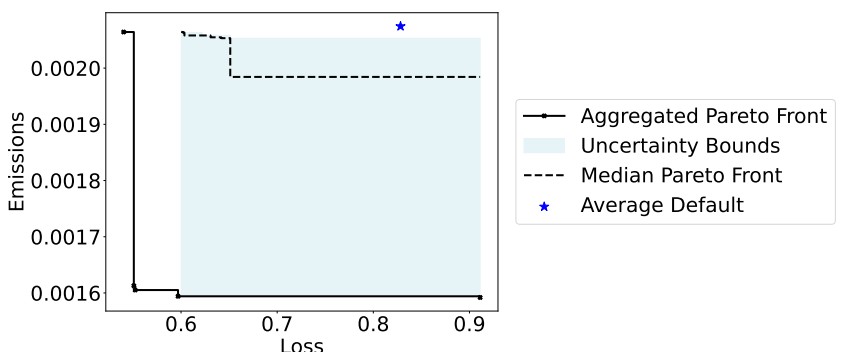

Figure 1: Pareto front for EfficientNetV2 on CIFAR10 over multiple seeds. We show the loss in % and the emissions in $kgCO_2eq$. The plots include median Pareto fronts with uncertainty bars, as well as an aggregated Pareto front of Pareto-optimal solutions across all runs. The star denotes the averaged performance of the default DSNN configuration.

validates the need for proper HPO tuning since we found better configurations that take the energy-efficient DSNNs a significant step further by improving their accuracy and energy consumption.

Additionally, we optimized an EfficientNetV2-based DSNN on CIFAR-10 to evaluate whether our method can yield further improvements when applied to an architecture already optimized for efficiency, thereby testing its robustness and generalizability. The results are shown in Figure 1 with the corresponding Pareto-optimal solutions in Table 10 in the appendix. Again, we were able to identify multiple configurations that surpass the default DSNN w.r.t. prediciton performance and emissions at inferece, using our MOMF approach. While EfficientNetV2 (Tan & Le, 2021) is inherently designed for optmizing for performance and efficiency, using compound scaling, the need for carefully tuning the quantization hyperparameters is apparent. Compared to the default EfficientNetV2, we were able to reduce emissions by more than 20%.

### 5.2.2 Hyperparameter Importances

Another crucial aspect is the analysis of hyperparameter importance to learn their influence on a model and lay the foundation of our DSNN design insights in the next subsection. We use DeepCAVE (Sass et al., 2022) for analyzing the Pareto front of our MFMO analysis in Figure 2f, and computing the hyperparameter importance using fANOVA (Hutter et al., 2014).

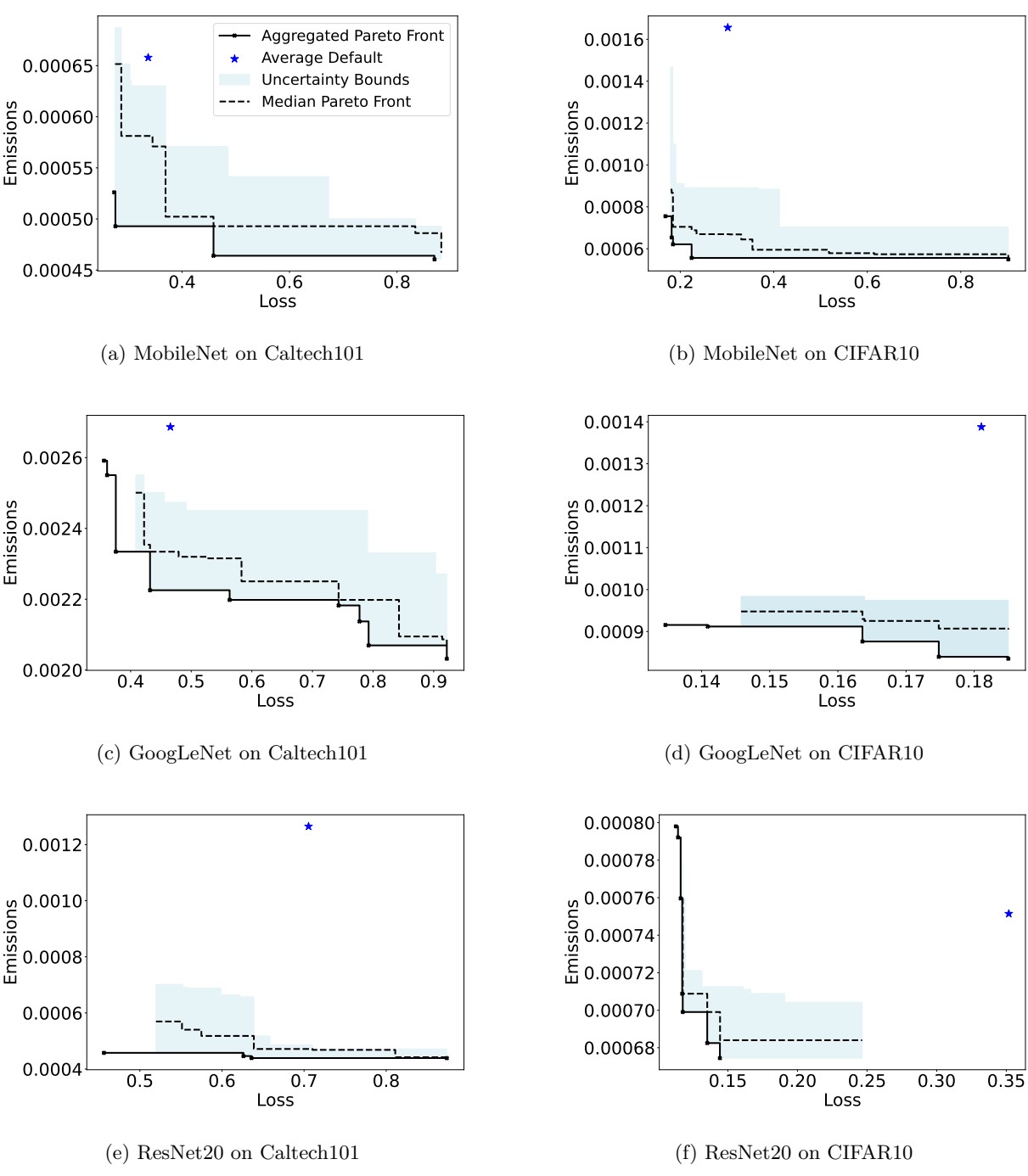

Figure 2: Comparison of Pareto fronts for MobileNet, GoogLeNet, and ResNet20 on Caltech101 and CIFAR10 datasets over multiple seeds. We show the loss in % and the emissions in kgCO$_2$eq. The plots include median Pareto fronts with uncertainty bars, as well as an aggregated Pareto front of Pareto-optimal solutions across all runs. The star denotes the averaged performance of the default DSNN configuration.

fANOVA fits a random forest surrogate model to the hyperparameter optimization landscape and decomposes the model's variance into components corresponding to hyperparameters. This allows fANOVA to estimate the marginal impact of individual hyperparameters or pairs of hyperparameters. For an extension to MO optimization, fANOVA can be applied to each objective's performance surface separately. The hyperparameter importances are then computed for different weightings of the objectives.

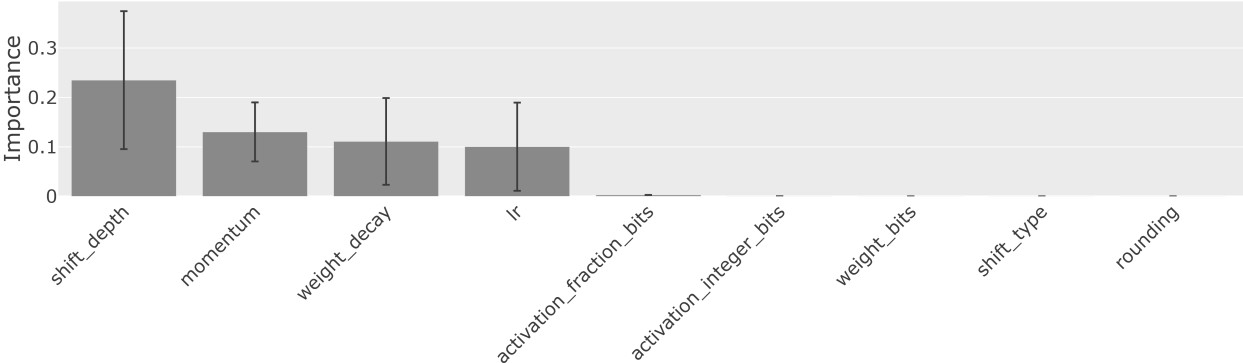

(a) Hyperparameter importance with respect to loss.

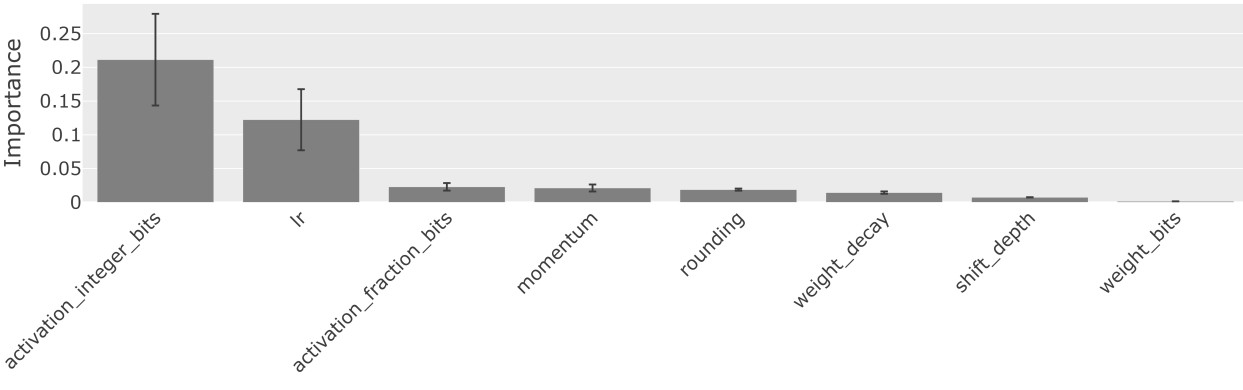

(b) Hyperparameter importance with respect to emissions.

Figure 3: Hyperparameter importance according to fANOVA for ResNet20 on CIFAR10. (a) Importance with respect to loss. (b) Importance with respect to emissions.

Figures 3a and 3b show the hyperparameter importances of a ResNet20 on Cifar10 w.r.t. loss and emissions, respectively. The MO-fANOVA analysis for different weightings of the objectives loss and emissions can be found in Figure 11 in the appendix. The most important DSNN-specific hyperparameters for emissions in Figure 3b include activation integer and fraction bits. This hints at the precision of the activation quantization being the most controlling factor for energy efficiency. Naturally, precision is a key factor since it controls the amount of operations in the network. Regarding loss in Figure 3a, the shift depth is the most important hyperparameter. The proportion of the network converted to perform shift operations naturally controls the amount of information retained in the network. This is crucial for the overall performance of the network.

Notably, the shift type has a low importance value in Figure 3a and is not among the most important hyperparameters in Figure 3b. This could indicate that the shift type is marginally relevant for both objectives. In practice, this insight could aid in model training by allocating fewer resources to tuning this hyperparameter, allocating resources tailored to the use case. This is further supported by looking at Tables

4 to 10. About 50% of the Pareto optimal configurations use either shift type, meaning they are not leaning toward either to maximize either objective.

With the analysis of hyperparameter importance in our study, we offer a baseline of hyperparameters to include for future training and inference purposes. Including only the significant ones is a promising way of further boosting the energy efficiency of the DSNNs and the optimization process (Probst et al., 2019). Additional plots showing the hyperparameter importance for each architecture can be found in Appendix A.2. DSNN-specific hyperparameters consistently place among the most influential, alongside weight decay and learning rate. This consistent importance underscores the relevance of understanding and optimizing DSNNs.

### 5.2.3 Insights into the Design of DSNNs

When looking at the configurations of the ResNet20 architecture on CIFAR10 in Table 4 (see Appendix for similar tables for the other architectures and datasets), most solutions have a surprisingly small shift depth $s \in \{1, 3\}$, compared to 20 as the maximal value and the setting of the default. At the same time, the number of activation fraction bits is often rather high. This leads to the assumption that the bulk of information is retained in the fraction part of the activation value. A valid expectation since we used batch normalization in the ResNet20, same as Elhoushi et al. (2021). In batch normalization, the layer inputs are re-scaled and re-centered using the mean and variance of the corresponding dimension (Ioffe & Szegedy, 2015). This usually leads to small weights, highlighting the importance of activation fraction precision, which is higher in Pareto optimal configurations. It ranges from 8 to 32, likely a contributing factor to the emission reduction.

While we initially hypothesized a direct proportionality between shift depth and emission savings, our analysis reveals a more nuanced relationship between hyperparameters, especially the shift depth and the bit precision in weights and activations. These hyperparameters interact in a non-linear manner, jointly influencing both energy consumption and model performance in ways that are not immediately intuitive. The results of our Pareto front analysis indicate that the choice of shift layers, which we expected to correlate positively with performance gains and negatively with loss, does not exhibit such a straightforward relationship. Instead, our findings highlight the intricate dependencies among architectural design choices, hyperparameter configurations, and their combined impact on energy efficiency. Notably, increasing the number of shift layers does not consistently result in greater emission savings. Weight precision also influences emissions, but no consistent trend was observed in the ResNet20 configurations, indicating the need for dataset-specific tuning.

These insights are corroborated by looking at the additional experiments in the appendix. Additionally to the Resnet20 on CIFAR10, we computed the Pareto fronts of our MOMF analysis on MobileNetV2 and GoogLeNet on CIFAR10. The corresponding results are shown in Figures 2b and 2d. The overall Pareto optimal configurations from multiple seeds can be found Appendix A.1 in Tables 5 and 6. Again, the shift depths are generally very low, either one or three, with two exceptions of seven and fourteen. The number of activation fraction bits is usually close to the upper bound of 32 bits.

We have also computed Pareto fronts of our MFMO approach on the Caltech101 dataset, for ResNet20, MobileNetV2 and GoogLeNet. The results can be seen in Figures 2a, 2c and 2e. The configurations on the Pareto fronts are detailed in Tables 7, 8 and 9. Again, the vast majority of configurations have a low shift depth in the range of $s \in \{1, 2, 3, 4, 5, 6\}$. Only two GoogLeNet configurations have a shift depth of eight and 9. This is still relatively low, given that GoogLeNet is a complex architecture with 22 layers in total. As with the previously discussed results, this contributes to the reduction of emissions while not impacting the performance. Generally, the number of activation fraction and integer bits increases with lower shift depth and vice versa. This confirms our previous findings from the thoroughly discussed ResNet20 on CIFAR10.

In Figures 3a and 3b, the learning rate is surprisingly identified as an important hyperparameter both for loss and emissions. To shed light on how the learning rate is related to emissions, we investigated this further. To this end, we instantiated a ResNet20 on CIFAR10 with the default configuration, but varied the learning by 100 values randomly sampled from the configuration search space. Analyzing the resulting emissions leads to a mean value of 0.00106 kgCO2eq, which exceeds the range of our Pareto fronts. The standard deviation is 0.00156, which is itself close to the mean value. This shows, indeed, that randomizing the learning rate introduced great variability in the energy measurements. As a control measurement to rule out other confounding factors, we measured the energy required for inference for a model instantiated with the default

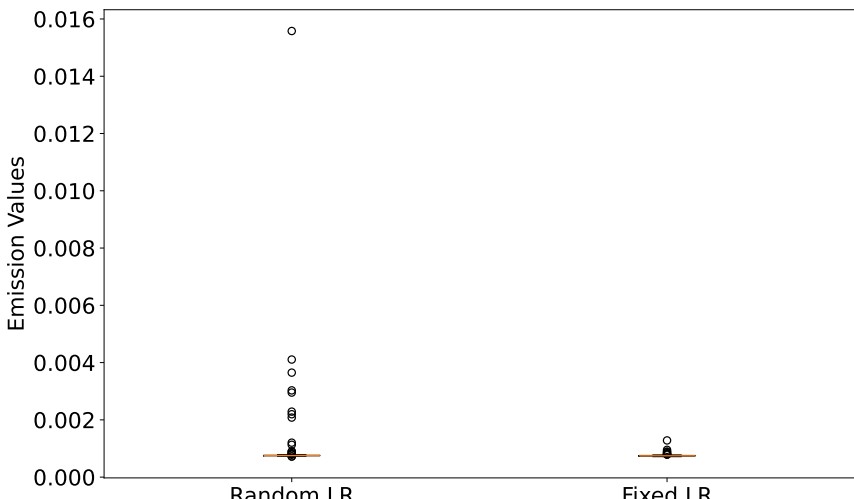

Figure 4: Emissions from different learning rates.

learning rate 100 times over. As expected, this leads to a mean of 0.00076 kgCo2eq, matching the evaluation of the default configuration in Figure 2f, and a very low standard deviation of about 0.00006 (see Figure 4).

This leads to the conclusion that the learning rate is indeed an influential HP on the emissions since it can drastically lead to an increase in emissions when chosen suboptimally. Our AutoML approach offers a solution to exactly this problem, since we aim to automate the identification and optimal choice of hyperparameters.

### 5.3 Transferability of Configurations

Understanding whether optimal configurations found in one setting remain effective in others is critical for practical and sustainable AutoML. In particular, the ability to transfer configurations across datasets can lead to substantial reductions in computational cost and associated emissions. If previously optimized configurations remain near-optimal despite changes in the data, users can avoid restarting the entire optimization process, thus saving both time and resources. To investigate this, we evaluated the Pareto-optimal configurations identified for a ResNet20 model on CIFAR-10 trained on ImageNet100, a subset of ImageNet Tian et al. (2020). We measure loss and emissions at inference. The results, shown in Figure 5 demonstrate that several of the original Pareto configurations remain competitive on Imagenet100, continuing to outperform the default configuration. Notably, the default remains Pareto-dominated while the transferred configurations yield improvements in both loss and emissions. This suggests a degree of robustness and transferability, which could be leveraged in practice to reduce the frequency of full re-optimization runs.

## 6 Conclusion

In this work, we presented our Green AutoML approach towards the sustainable optimization of DSNNs through a multi-fidelity, multi-objective (MFMO) HPO framework. Our approach effectively addressed the critical intersection between advancing the capabilities of Deep Learning and environmental sustainability. By leveraging AutoML tools and integrating the environmental impact as an objective, we adeptly navigated the trade-off between model performance and efficient resource utilization. Our experimental results focused on a better understanding of DSNNs. We successfully optimized DSNNs to achieve higher accuracy while minimizing energy consumption, surpassing the default configuration settings in both aspects. Through systematic experimentation, we identified key hyperparameters that significantly influence performance and emissions, such as shift depth and number of weight bits.

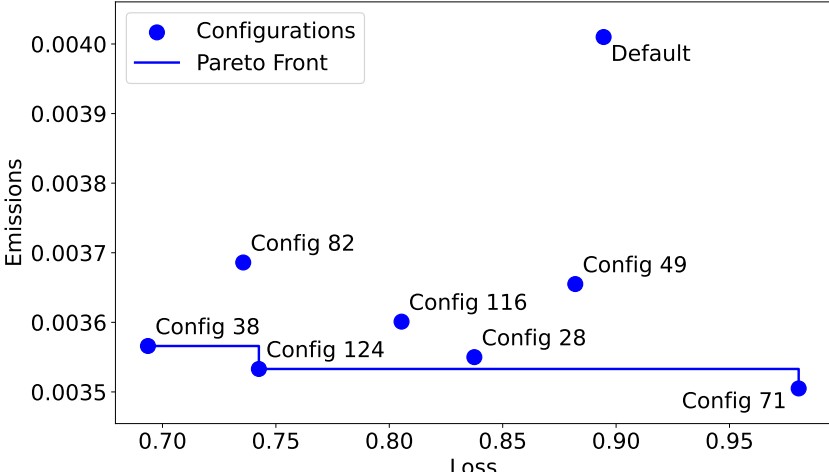

Figure 5: ResNet20 trained on Imagenet100, instantiated with Pareto-optimal configurations from CIFAR10 (see Table 4). We show the loss in % and the emissions in kgCO$_2$eq.

By optimizing these hyperparameters, our MFMO approach did not just improve one dimension of the problem – it concurrently enhanced both model loss and energy efficiency, showcasing a balanced improvement across essential performance metrics. We have thoroughly explored the configuration space for DSNNs, introduced a Green AutoML approach for efficiency-driven model development (Tornede et al., 2023), and provided valuable insights into the design decisions impacting DSNN performance.

The generalizable insights from our work extend beyond the need to carefully select bitwidths for quantization. Our results reveal that optimal configurations for DSNNs are often counterintuitive and highly dependent on the intricate relations between hyperparameters. For example, we found that low shift depths often achieve superior trade-offs between accuracy and energy efficiency, challenging assumptions about full quantization of networks. Additionally, our analysis highlights the importance of prioritizing specific hyperparameters for different objectives, providing a targeted approach to DSNN optimization. These findings are consistent across multiple architectures and datasets, demonstrating their broader applicability. Overall, our study shows the need for a systematic, automated approach to detecting these insights, which are relevant not only for DSNNs but also for other quantized and resource-efficient neural networks.

**Limitations** While our use of CodeCarbon provides valuable insights into the emissions impact of training and inference for DSNNs, it is important to acknowledge the limitations and assumptions inherent in these measurements. CodeCarbon relies on real-time power draw metrics from tools like nvidia-smi. However, these estimates assume a steady power draw during computation and do not account for fluctuations in hardware utilization or dynamic changes in the energy grid. We consider CodeCarbon a reasonable approximation for measuring energy consumption. This is reflected in recent literature, where studies compared CodeCarbon measurements to wattmeters that directly measure power consumption on machines (Bouza et al., 2023).

We acknowledge that energy consumption varies significantly based on regional energy mixes and the infrastructure of data centers. Emissions are typically calculated by multiplying the energy consumed by a carbon intensity factor. For a given energy mix, this factor remains constant. As a result, emissions scale linearly with energy consumption when the energy mix is fixed. Thus, once we compute a Pareto front based on energy consumption, the emission axis can be scaled accordingly to reflect the specific carbon intensity of the region. In dynamic scenarios where energy mixes change over time, our approach enables rapid recalculation of Pareto fronts to reflect these variations so that stakeholders can accommodate real-time or regional changes in energy conditions. While data centers may often rely on carbon-neutral energy sources, the broader applicability of our method extends to embedded devices, such as those used in automobiles

or IoT systems. These devices operate in highly varied environments, with energy sources dependent on local conditions. Our approach enables stakeholders to determine location-specific or even time-sensitive Pareto fronts, providing a tailored basis for decision-making. This empowers users to balance trade-offs between accuracy and emissions based on financial, ecological, or operational considerations. While testing on hardware such as low-power chips or alternative GPUs could offer additional insights into hardware-specific performance trade-offs, we maximize the potential impact and accessibility of our research by training on NVIDIA A100 GPUs, which is commonly-used hardware in academia and industry.

The DSNN configurations that outperform the default baselines were identified using our MFMO approach. This strategy enabled efficient exploration of a broader region of the search space while keeping training costs manageable. Importantly, the models were not trained to full convergence, by design. Although we expect the accuracy gap between configurations to narrow with full training, our optimal configurations consistently demonstrate higher accuracy in the early stages of training, indicating faster convergence. This approximation — inferring full-training behavior from early-stage performance — is a widely accepted assumption in multi-fidelity optimization. The early stage performance advantage is particularly valuable during optimization, as it enables the rapid identification of promising candidates with significantly reduced computational overhead. Crucially, we also expect the relative emissions savings of these configurations to persist after full training, since the computational graph, operation types, and numerical precision remain fixed throughout training and thus continue to govern energy consumption consistently.

We do not focus on transformer-based architectures in this work. While transformers have become prominent in computer vision, particularly Vision Transformers (ViTs), they typically require substantial computational resources and are less common in highly constrained environments. In scenarios such as edge computing and embedded systems for image classification, smaller convolutional architectures remain a predominant choice due to their lower memory footprint, faster inference, and mature optimization toolchains (Maurício et al., 2023). Although there are emerging quantization strategies for transformers (Scherer et al., 2024), our goal is not to replicate or extend the DSNN framework to support these architectures. In line with the principles of AutoML, we are conscious of the compute budget required for large-scale search and deliberately avoid high-cost models like transformers to ensure that our pipeline aligns with low-resource deployment scenarios.

Weighing the additional cost of employing AutoML with the savings achieved as a result is vital to assessing the impact of our approach. We argue that we offset the additional cost of the AutoML training process, namely the generation of the Pareto front for each model, through our savings in inference cost in a negligible amount of time. We target an application environment like automated driving or automated production, where inference needs to happen at near real-time or real-time. To provide sufficient evidence, we provide an upper bound calculation based on the experiments we ran. Our longest AutoML optimization run, GoogLeNet on Caltech101, took approximately 48 hours to determine an approximated Pareto front, which is by far the longest; most other runs were completed in about half that time. Assuming a maximum 300W power draw on an A100 GPU according to NVIDIA and a local carbon intensity factor of 0.475 kgCO2/kWh, this results in an estimated emission of about 6.84 kgCO2eq, based on the CodeCarbon formula:

$$\text{Emissions (kgCO2eq)} = 0.3\,\text{kW} * 48\,\text{h} * 0.475\,\text{kgCO2/kWh} \approx 6.84\,\text{kgCO2eq}. \tag{6.1}$$

For GoogLeNet, our optimized model reduces inference emissions by approximately 0.0004 kgCO2eq per inference, meaning the optimization overhead is amortized after around 17,100 inferences, or about 4.75 hours assuming a conservative real-time inference rate of one image per second. This is much slower than actual rates in automated driving or production contexts, which is what we are targeting. For other models in our study, per-inference emission savings are significantly higher, reaching the break-even point even earlier.

In this work, we focus exclusively on image classification tasks. Other vision tasks, such as object detection, semantic segmentation, or visual tracking, are relevant for energy-efficient computing in real-world applications, particularly in our intended domains like automated driving and industrial monitoring. However, extending our methodology to these more complex tasks is beyond the scope of this study.

**Future Work** could focus on revisiting the multi-fidelity-multi-objective implementation to find a more efficient way for ParEGO and HyperBand to intertwine, such as by finding a more effective way to assign

budgets and weights of objectives. Further investigations could include exploring more DSNN-specific fidelity types and multi-objective algorithms to achieve even greater reductions in model emissions. We consider it especially interesting to use the number of weight bits as a fidelity type. By controlling the precision of the weight quantization, training can be sped up in the earlier fidelity while regaining as much information as possible, to use this for full training of the most promising configurations at maximum precision. Through these future initiatives, the underlying optimization methodology and the environmental benefits of our optimized DSNNs can be further improved, thereby contributing significantly to sustainable AI.

## Acknowledgements

The authors were supported by the German Federal Ministry for the Environment, Climate Action, Nature Conservation and Nuclear Safety (BMUKN) (GreenAutoML4FAS project no. 67KI32007A). The authors gratefully acknowledge the computing time provided to them on the high-performance computers Noctua2 at the NHR Center PC2 under the project hpc-prf-intexml. These are funded by the Federal Ministry of Education and Research and the state governments participating on the basis of the resolutions of the GWK for the national high performance computing at universities (www.nhr-verein.de/unsere-partner).

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

# A Appendix

## A.1 Optimization Results

We provide tables that list the Pareto-optimal configurations obtained for ResNet20, MobileNet, GoogLeNet and EfficientNetV2 on CIFAR-10 and Caltech101 (Tables 4 to 10). For each model and dataset, we report the mean and standard deviation across multiple seeds. These results include both the Pareto-optimal solutions and the performance of the default configurations in terms of loss and emissions, offering a comprehensive overview of the trade-offs achieved.

## A.2 Hyperparameter Importances

We further analyzed the hyperparameter importance for the additional architectures we examined, as shown in Figures 6a to 10b. Across all models, the DSNN-specific hyperparameters consistently remain among the most important ones, alongside weight decay and learning rate. This highlights the relevance of our approach: focusing on the DSNN-specific parameters and allocating computational resources toward tuning them optimally for the given task appears to be a promising and effective strategy.

## A.3 Additional Baselines for Comparison

To demonstrate the effectiveness and relevance of our approach, we compare against the same baselines as the original DSNN paper by Elhoushi et al. (2021). Since our work builds on a multi-objective optimization framework targeting both accuracy and emissions, but the original DSNN paper provides only single-objective baselines (focused solely on accuracy), we restricted our evaluation to the accuracy objective for fair comparison. Specifically, we focus on the VGG19 architecture (Simonyan & Zisserman, 2014), as this is the network for which Elhoushi et al. (2021) published baseline results.

From the Pareto front obtained via MOMF optimization of the VGG19 architecture, we selected a Pareto-optimal configuration that outperformed the default in accuracy while maintaining similar emissions. As shown in Table 2, the final accuracy of this configuration outperforms not only the original unquantized VGG19 but also the quantized network baselines such as the DeepShift-PS baseline from Elhoushi et al. (2021), AdderNet (Chen et al., 2020b), and ShiftAddNet (You et al., 2020).

|  | Model | Accuracy |
|---|---|---|
|  | **AutoDSNN** | **93.45%** |
|  | Original* | 92% |
| **VGG19 on CIFAR10** | DeepShift-PS* | 91.57% |
|  | AdderNet* (Chen et al., 2020b) | 93.02% |
|  | ShiftAddNet* (You et al., 2020) | 90% |

Table 2: Accuracy comparison of VGG19 model variants on CIFAR10. Baseline results from Elhoushi et al. (2021) are marked with an asterisk.

Looking at the configuration details in Table 3, we can see that the Pareto optimal configuration that beats the aforementioned baselines has 11 shift layers, meaning that little more than half of the VGG19 layers are quantized. This is in line with our analysis of optimal DSNN configurations in Section 5.2.3. Again, it is not optimal to simply quantize the whole architecture. More so, since with our MOMF approach we can beat both the original unquantized architecture, the default DSNN, as well as both quantization baselines. These results highlight the relevance of our approach to discovering well-performing DeepShift configurations tailored to both accuracy and emissions. In particular, our optimal configuration, which outperforms all considered baselines, is only partially quantized, demonstrating that effective configurations are not straightforward or easily designed by hand. This underscores the importance of our method for systematically identifying efficient DSNNs beyond standard quantization heuristics, which are competitive with relevant baselines.

Table 3: Configuration for VGG19 on CIFAR10, selected from the Pareto front. This configuration achieves a significant improvement over the default unquantized VGG19 in terms of accuracy.

| Hyperparameter | Config 32 |
|---|---|
| Optimizer | Adam |
| Learning Rate | 0.0107 |
| Momentum | 0.1474 |
| Weight Decay | 0.0032 |
| Weight Bits | 5 |
| Act. Int. Bits | 20 |
| Act. Frac. Bits | 11 |
| Shift Depth | 11 |
| Shift Type | Q |
| Rounding | Stochastic |

Table 4: Pareto optimal configurations and default DSNN initiation of ResNet 20 on CIFAR10. Pareto optimal solutions on the aggregated Pareto front of the ResNet20 architecture on CIFAR10 on three seeds, including the mean aggregated loss and emissions of the default configuration.

| Hyperpar. | Config 124 | Config 116 | Config 71 | Config 38 | Config 82 | Config 28 | Config 49 | Default |
|---|---|---|---|---|---|---|---|---|
| Optimizer | Ranger | Ranger | Ranger | Ranger | Ranger | Adagrad | Adagrad | SGD |
| Learning Rate | 0.0130 | 0.0327 | 0.0150 | 0.0542 | 0.0129 | 0.0548 | 0.0929 | 0.1 |
| Momentum | 0.7489 | 0.7718 | 0.1529 | 0.4983 | 0.6838 | 0.6783 | 0.4825 | 0.9 |
| Weight Decay | 0.0001 | 0.00005 | 0.0038 | 0.0001 | 0.0001 | 0.0002 | 0.0003 | 0.0001 |
| Weight Bits | 2 | 8 | 5 | 2 | 2 | 5 | 5 | 5 |
| Act. Integer Bits | 11 | 13 | 2 | 9 | 11 | 22 | 24 | 16 |
| Act. Fraction Bits | 32 | 30 | 32 | 11 | 27 | 11 | 8 | 16 |
| Shift Depth | 1 | 1 | 3 | 1 | 1 | 1 | 1 | 20 |
| Shift Type | PS | PS | Q | PS | PS | PS | PS | PS |
| Rounding | Det. | Det. | Det. | Stochastic | Det. | Stochastic | Stochastic | Det. |
| **Loss** | 0.1127 | 0.1142 | 0.1161 | 0.1172 | 0.1176 | 0.1352 | 0.1443 | 0.3518 |
| **Emissions** | 0.000798 | 0.000792 | 0.000759 | 0.000708 | 0.000699 | 0.000682 | 0.000674 | 0.000751 |

Table 5: Pareto optimal configurations and default DSNN instantiation of MobileNetV2 on CIFAR10

| Hyperparameter | Config 15 | Config 68 | Config 66 | Config 21 | Default |
|---|---|---|---|---|---|
| Optimizer | Adadelta | Adadelta | Adadelta | SGD | SGD |
| Learning Rate | 0.182188 | 0.186665 | 0.197817 | 0.183219 | 0.1 |
| Momentum | 0.726835 | 0.689596 | 0.1837 | 0.76337 | 0.9 |
| Weight Decay | 0.002727 | 0.003048 | 0.00306 | 0.002277 | 0.0001 |
| Weight Bits | 5 | 5 | 5 | 4 | 5 |
| Activation Integer Bits | 21 | 19 | 21 | 23 | 16 |
| Activation Fraction Bits | 31 | 31 | 32 | 16 | 16 |
| Shift Depth | 14 | 7 | 1 | 1 | 53 |
| Shift Type | PS | PS | PS | PS | PS |
| Rounding | Deterministic | Deterministic | Deterministic | Deterministic | Deterministic |
| **Loss** | 0.1683 | 0.1756 | 0.1797 | 0.9016 | 0.3017 |
| **Emissions** | 0.000755 | 0.000655 | 0.000552 | 0.000549 | 0.001656 |

Table 6: Pareto optimal configurations and default DSNN instantiation of GoogLeNet on CIFAR10

| Hyperpar. | Config 65 | Config 33 | Config 28 | Config 14 | Config 32 | Default |
|---|---|---|---|---|---|---|
| Optimizer | Adadelta | Adadelta | Ranger | Ranger | Adadelta | SGD |
| Learning Rate | 0.028997 | 0.023838 | 0.020002 | 0.058610 | 0.115941 | 0.1 |
| Momentum | 0.209328 | 0.494258 | 0.250184 | 0.673880 | 0.372339 | 0.9 |
| Weight Decay | 0.008487 | 0.006737 | 0.006691 | 0.002313 | 0.009464 | 0.0001 |
| Weight Bits | 2 | 3 | 2 | 4 | 2 | 5 |
| Activation Integer Bits | 21 | 24 | 29 | 26 | 29 | 16 |
| Activation Fraction Bits | 4 | 5 | 4 | 8 | 7 | 16 |
| Shift Depth | 1 | 1 | 1 | 1 | 1 | 22 |
| Shift Type | Q | Q | Q | PS | PS | PS |
| Rounding | Stochastic | Deterministic | Deterministic | Deterministic | Deterministic | Deterministic |
| **Loss** | 0.1347 | 0.1409 | 0.1636 | 0.1748 | 0.1850 | 0.1810 |
| **Emissions** | 0.000916 | 0.000912 | 0.000876 | 0.000840 | 0.000835 | 0.001388 |

Table 7: Pareto optimal configurations and default DSNN instantiation of ResNet20 on Caltech101

| Hyperparameter | Config 26 | Config 66 | Config 32 | Config 76 | Default |
|---|---|---|---|---|---|
| Optimizer | Ranger | Ranger | Ranger | RMSProp | SGD |
| Learning Rate | 0.023 | 0.039 | 0.076 | 0.015 | 0.1 |
| Momentum | 0.559 | 0.333 | 0.344 | 0.551 | 0.9 |
| Weight Decay | 0.0033 | 0.0027 | 0.0029 | 0.0069 | 0.0001 |
| Weight Bits | 2 | 5 | 2 | 5 | 5 |
| Activation Integer Bits | 24 | 21 | 22 | 25 | 16 |
| Activation Fraction Bits | 17 | 27 | 13 | 21 | 16 |
| Shift Depth | 2 | 2 | 1 | 1 | 20 |
| Shift Type | PS | PS | PS | Q | PS |
| Rounding | Deterministic | Stochastic | Stochastic | Deterministic | Deterministic |
| **Loss** | 0.456 | 0.532 | 0.636 | 0.874 | 0.679 |
| **Emissions** | 0.00046 | 0.00044 | 0.00044 | 0.00044 | 0.00109 |

Table 8: Pareto optimal configurations and default DSNN instantiation of MobileNetV2 on Caltech101

| Hyperparameter | Config 9 | Config 65 | Config 33 | Config 74 | Default |
|---|---|---|---|---|---|
| Optimizer | Adadelta | Adadelta | SGD | Adam | SGD |
| Learning Rate | 0.192 | 0.199 | 0.004 | 0.006 | 0.1 |
| Momentum | 0.510 | 0.545 | 0.016 | 0.012 | 0.9 |
| Weight Decay | 0.009 | 0.004 | 0.004 | 0.004 | 0.0001 |
| Weight Bits | 3 | 2 | 5 | 4 | 5 |
| Activation Integer Bits | 8 | 12 | 11 | 32 | 16 |
| Activation Fraction Bits | 26 | 30 | 7 | 6 | 16 |
| Shift Depth | 6 | 3 | 1 | 1 | 53 |
| Shift Type | PS | PS | Q | Q | PS |
| Rounding | Deterministic | Deterministic | Deterministic | Stochastic | Deterministic |
| **Loss** | 0.274 | 0.276 | 0.459 | 0.870 | 0.337 |
| **Emissions** | 0.00053 | 0.00049 | 0.00046 | 0.00046 | 0.00066 |

Table 9: Pareto optimal configurations and default DSNN instantiation of GoogLeNet on Caltech101

| Hyperpar. | Config 66 | Config 25 | Config 44 | Config 63 | Config 33 | Config 21 | Config 20 | Default |
|---|---|---|---|---|---|---|---|---|
| Optimizer | Adadelta | Adadelta | Adadelta | Adagrad | Radam | Adagrad | Adam | SGD |
| Learning Rate | 0.058 | 0.059 | 0.052 | 0.13 | 0.199 | 0.187 | 0.017 | 0.1 |
| Momentum | 0.185 | 0.642 | 0.194 | 0.889 | 0.647 | 0.367 | 0.248 | 0.9 |
| Weight Decay | 0.0004 | 0.0019 | 0.0029 | 0.0006 | 0.0027 | 0.0048 | 0.0059 | 0.0001 |
| Weight Bits | 3 | 4 | 3 | 2 | 4 | 3 | 2 | 5 |
| Activation Integer Bits | 25 | 26 | 9 | 10 | 7 | 30 | 7 | 16 |
| Activation Fraction Bits | 5 | 8 | 19 | 7 | 20 | 4 | 30 | 16 |
| Shift Depth | 8 | 1 | 9 | 1 | 2 | 2 | 2 | 22 |
| Shift Type | Q | PS | PS | PS | PS | Q | Q | PS |
| Rounding | Det. | Det. | Det. | Det. | Det. | Stochastic | Stochastic | Det. |
| **Loss** | 0.356 | 0.362 | 0.376 | 0.563 | 0.778 | 0.793 | 0.922 | 0.466 |
| **Emissions** | 0.0026 | 0.0026 | 0.0023 | 0.0022 | 0.0021 | 0.0021 | 0.0020 | 0.0027 |

Table 10: Pareto optimal configurations and default DSNN instantiation of EfficientNetV2 on CIFAR10

| Hyperpar. | Config 61 | Config 75 | Config 31 | Config 71 | Config 39 | Default |
|---|---|---|---|---|---|---|
| Optimizer | RMSprop | RMSprop | Adadelta | Adam | Ranger | SGD |
| Learning Rate | 0.01737 | 0.03561 | 0.03508 | 0.17490 | 0.12583 | 0.1 |
| Momentum | 0.8861 | 0.6085 | 0.4682 | 0.3632 | 0.7668 | 0.9 |
| Weight Decay | 0.0000381 | 0.007015 | 0.003031 | 0.009879 | 0.007242 | 0.0001 |
| Weight Bits | 5 | 2 | 3 | 4 | 2 | 5 |
| Activation Integer Bits | 8 | 15 | 22 | 16 | 9 | 16 |
| Activation Fraction Bits | 11 | 21 | 22 | 18 | 10 | 16 |
| Shift Depth | 1 | 51 | 48 | 28 | 42 | 342 |
| Shift Type | Q | PS | Q | PS | PS | PS |
| Rounding | Stochastic | Deterministic | Stochastic | Deterministic | Stochastic | Deterministic |
| **Loss** | 0.1521 | 0.1539 | 0.1555 | 0.1653 | 0.1657 | 0.8283 |
| **Emissions** | 0.000973 | 0.000934 | 0.000926 | 0.000880 | 0.000870 | 0.002074 |

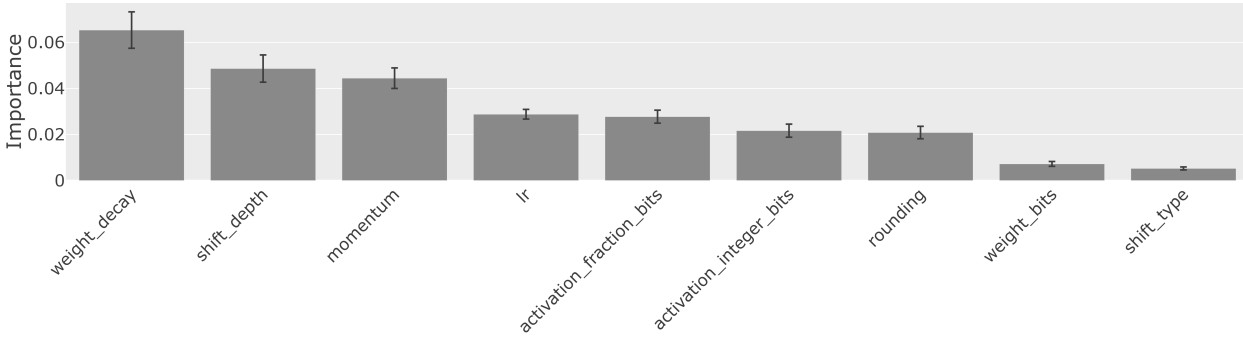

(a) Hyperparameter importance with respect to loss.

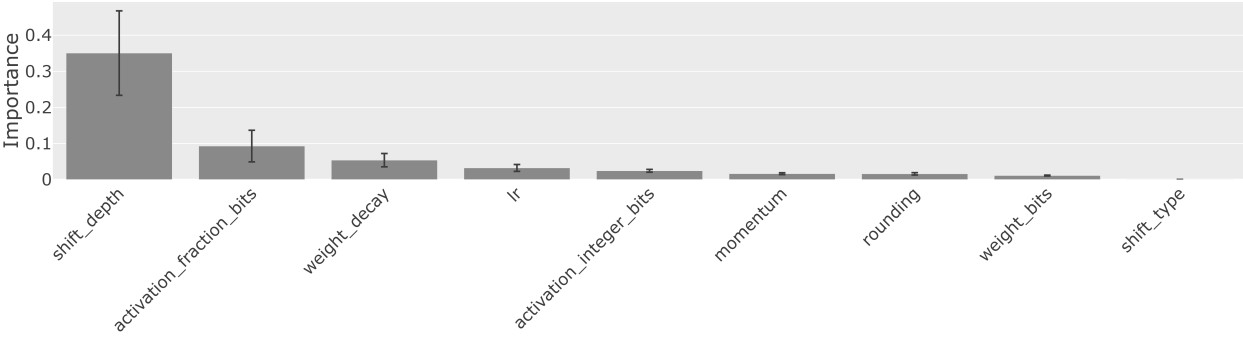

(b) Hyperparameter importance with respect to emissions.

Figure 6: Hyperparameter importance according to fANOVA for MobileNet on Caltech101. (a) Importance with respect to loss. (b) Importance with respect to emissions.

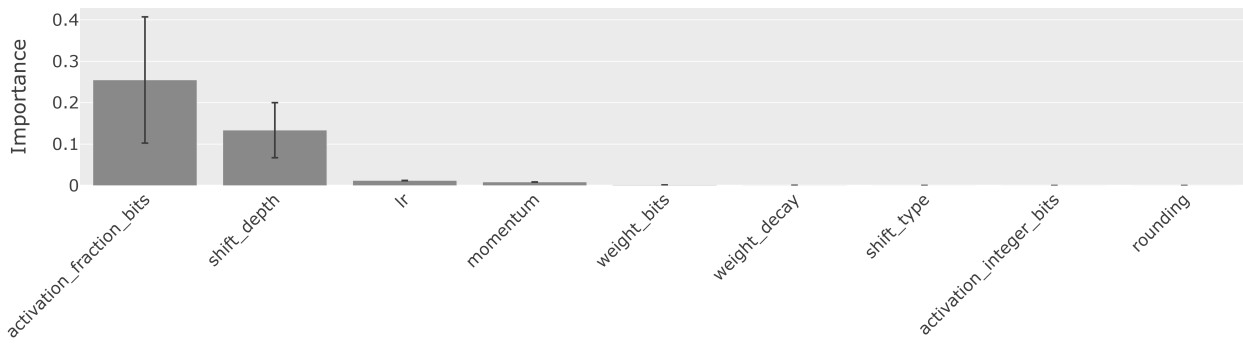

(a) Hyperparameter importance with respect to loss.

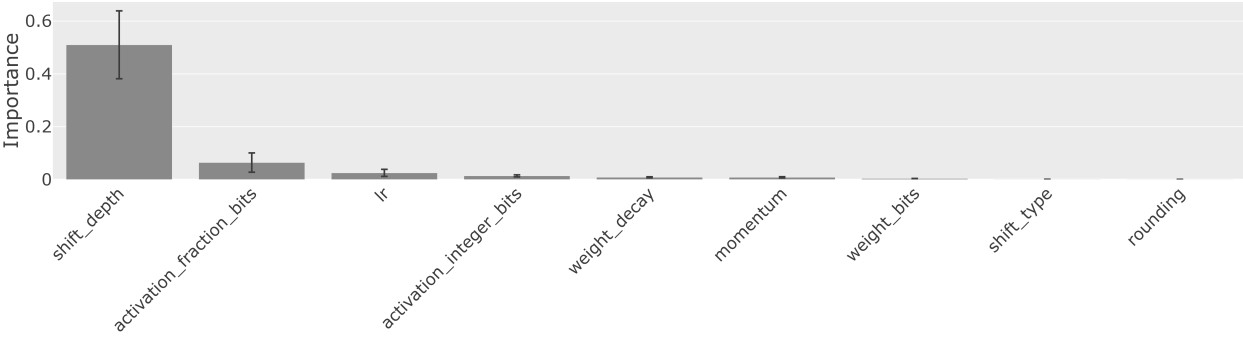

(b) Hyperparameter importance with respect to emissions.

Figure 7: Hyperparameter importance according to fANOVA for GoogLeNet on Caltech101. (a) Importance with respect to loss. (b) Importance with respect to emissions.

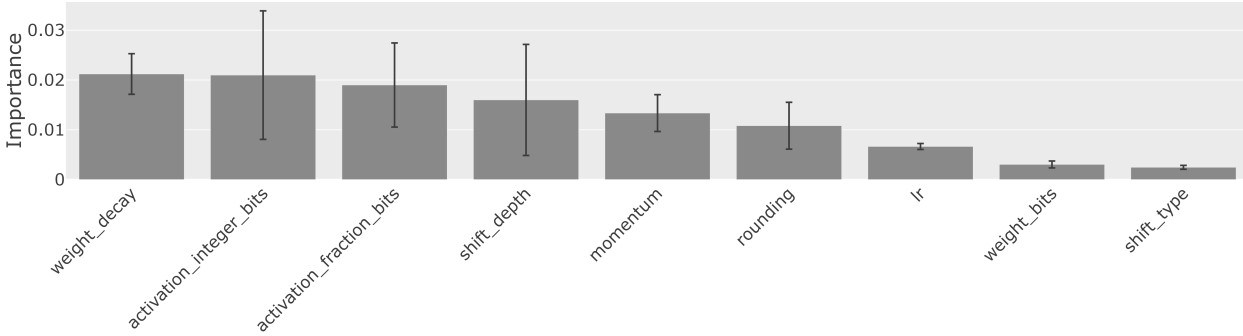

(a) Hyperparameter importance with respect to loss.

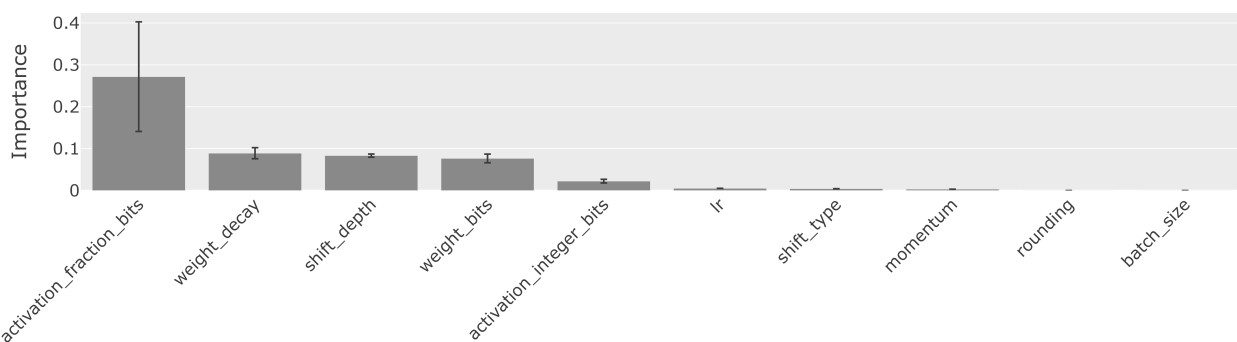

(b) Hyperparameter importance with respect to emissions.

Figure 8: Hyperparameter importance according to fANOVA for ResNet20 on Caltech101. (a) Importance with respect to loss. (b) Importance with respect to emissions.

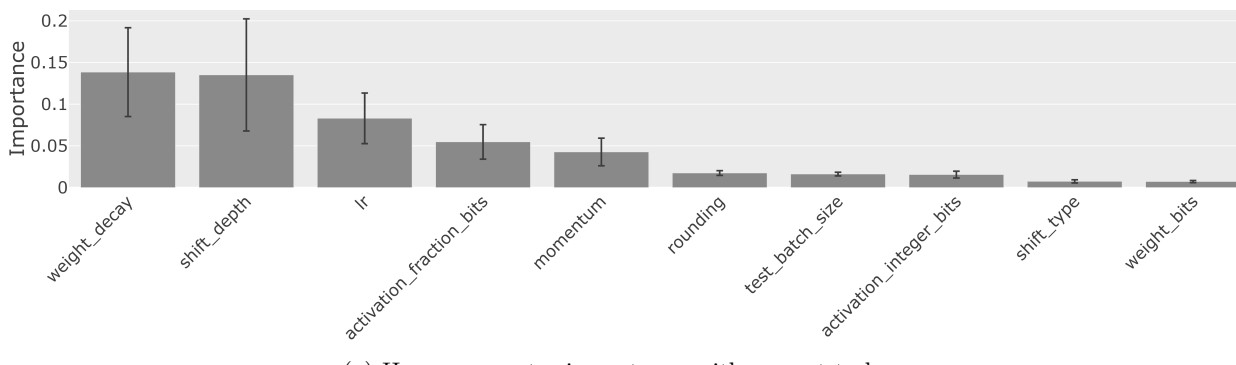

(a) Hyperparameter importance with respect to loss.

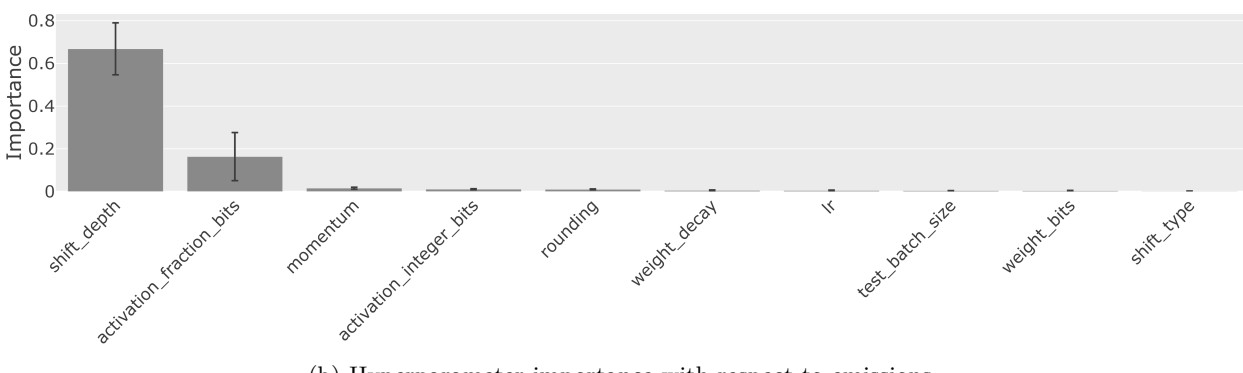

(b) Hyperparameter importance with respect to emissions.

Figure 9: Hyperparameter importance according to fANOVA for MobileNet on CIFAR10. (a) Importance with respect to loss. (b) Importance with respect to emissions.

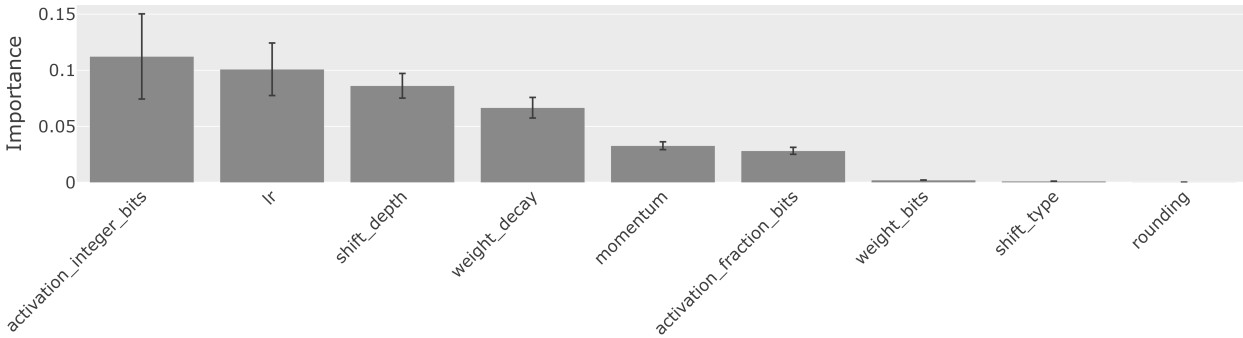

(a) Hyperparameter importance with respect to loss.

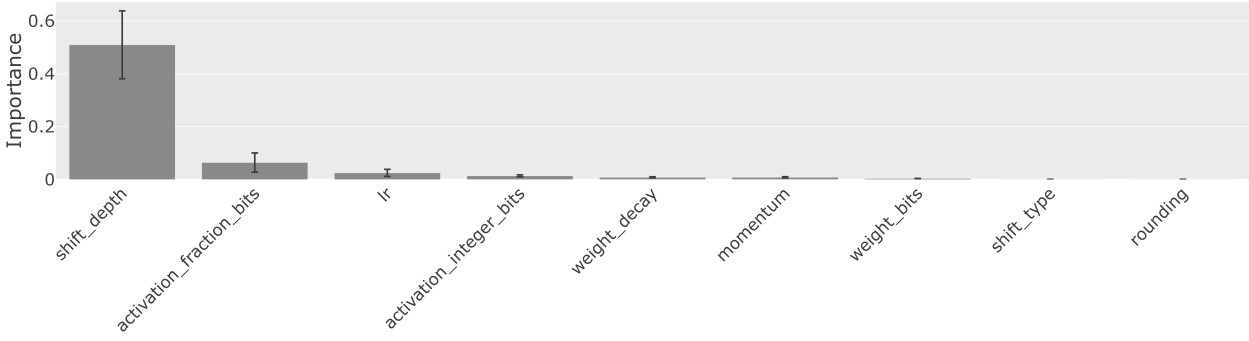

(b) Hyperparameter importance with respect to emissions.

Figure 10: Hyperparameter importance according to fANOVA for GoogLeNet on CIFAR10. (a) Importance with respect to loss. (b) Importance with respect to emissions.

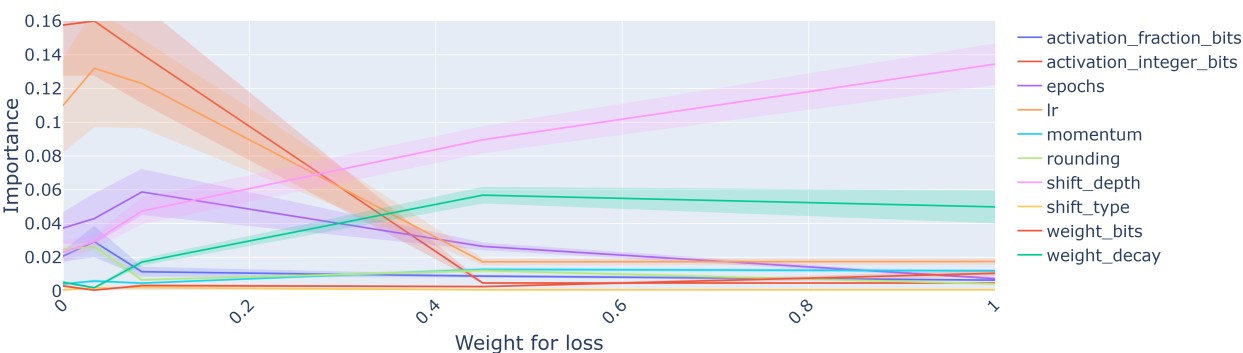

Figure 11: MO analysis of hyperparameter importance of a ResNet20 on CIFAR10 w.r.t. loss and emissions using the fANOVA method. The x-axis shows $w_l$, the weight of the objective loss, ranging from 0 to 1. The weight of the objective emissions is thus $w_e = 1 - w_l$. The y-axis shows the importance of each hyperparameter in the legend.

---

**Algorithm 1** Multi-Fidelity Optimization with Parego for DSNNs

---

**Require:** Configuration space $\mathcal{C}$, objectives $\mathcal{L} = [\mathcal{L}_{\text{loss}}, \mathcal{L}_{\text{emissions}}]$, budget range $[B_{\min}, B_{\max}]$, number of trials $N$, DSNN architecture $\mathcal{A}$.

**Ensure:** Pareto optimal configurations $\mathcal{P}_N$.

1: Initialize scenario $\mathcal{S}$ with $\mathcal{C}$, $\mathcal{L}$, $(B_{\min}, B_{\max})$, and $N$.
2: Generate initial observation dataset $\mathcal{D}_{\text{init}}$ by sampling $k$ random configurations $\{\lambda_i\}_{i=1}^k \subset \mathcal{C}$.
3: Define intensifier $\mathcal{H}$ as Hyperband for budget allocation.
4: Initialize optimizer $\mathcal{O}$ using Parego and $\mathcal{H}$.
5: **for** each trial $t \in \{1, \ldots, N\}$ **do**
6:     Select a configuration $\lambda_t \in \mathcal{C}$ using Parego.
7:     Allocate budget $b_t \in (B_{\min}, B_{\max})$ using $\mathcal{H}$.
8:     Perform evaluation of $\lambda_t$ with budget $b_t$:

    1. Convert DSNN $\mathcal{A}$ to a shift-based architecture $\mathcal{A}'$:

$$\mathcal{A}' = \text{convert\_to\_shift}(\mathcal{A}, \lambda_t[\text{shift\_depth}], \lambda_t[\text{shift\_type}])$$
$$w' = \text{round}(w, \lambda_t[\text{rounding}]), \quad \forall w \in \mathcal{A}',$$

    where convert\_to\_shift replaces standard operations with shift operations, and round applies deterministic or stochastic rounding.

    2. Train $\mathcal{A}'$ for $b_t$ epochs and compute the objective values:

$$\mathcal{L}_{\text{loss}}(\lambda_t) = \frac{1}{|D_{\text{test}}|} \sum_{(x,y) \in D_{\text{test}}} \ell(f_{\mathcal{A}'}(x), y),$$
$$\mathcal{L}_{\text{emissions}}(\lambda_t) = \text{measure\_emissions}(\mathcal{A}', b_t),$$

    where $D_{\text{test}}$ is the test dataset, $\ell$ is the cross-entropy loss, and measure\_emissions computes the energy consumption.

    3. Update observation dataset:

$$\mathcal{D}_t \leftarrow \mathcal{D}_{t-1} \cup \{(\lambda_t, [\mathcal{L}_{\text{loss}}(\lambda_t), \mathcal{L}_{\text{emissions}}(\lambda_t)])\}.$$

9:     Update the Pareto front:

$$\mathcal{P}_t = \{(\lambda, \mathcal{L}(\lambda)) \in \mathcal{D}_t \mid \nexists \lambda' \in \mathcal{D}_t : \mathcal{L}(\lambda') \succ \mathcal{L}(\lambda)\},$$

    where $\mathcal{L}(\lambda') \succ \mathcal{L}(\lambda)$ denotes that $\lambda'$ dominates $\lambda$.
10: **end for**
11: **return** $\mathcal{P}_N$

---

