# OpenReview forum: "Leveraging AutoML for Sustainable Deep Learning: A Multi- Objective HPO Approach on Deep Shift Neural Networks"
_TMLR — Accepted by TMLR_

### Review · Reviewer_8oNe · 2025-04-08

**Summary Of Contributions:**

This paper proposes to apply Automated Machine Learning (AutoML) techniques such as Hyperparameter Optimization (HPO) to Deep Shifting Neural Networks (DSNN). DSNN operate using bit-shift operations as opposed to the more hardware-intensive multiply-add accumulate (MAC) operations. Specifically, the authors utilize Bayesian Optimization (BO) HPO techniques to this end, to optimize networks like ResNet-20 and MobileNetV2 on an Nvidia A100 GPU. Additionally, the authors provide insights into the most important DSNN design choices.

**Audience:**

Yes

**Broader Impact Concerns:**

N/A

References:

[1] Xiao, Muhan, et al. "RetNAS: Reinforce Evolutionary Trend in Lightweight Neural Architecture Search." 2024 IEEE Smart World Congress (SWC). IEEE, 2024.

[2] Wightman, Ross, Hugo Touvron, and Hervé Jégou. "Resnet strikes back: An improved training procedure in timm." arXiv preprint arXiv:2110.00476 (2021).

**Claims And Evidence:**

Yes

**Requested Changes:**

Experimental - the authors need to do at least one of two things:
1) Use larger and more powerful models that fully take advantage of the A100 GPU and where optimizations would be more meaningful. For instance, aim for something that would not fit on the A100's predecessor, the 32GB V100. Using an A100 for ResNet-20 and MobileNetV2 is very much overkill and subtracts from the substance of the paper's findings.
2) Experiment with smaller GPUs, potentially NPUs, ASICS, etc. for mobile or battery powered devices where energy savings are a bigger concern and where larger models cannot run.

Additionally, there should be some comparison with existing methods in the literature in these scenarios. The premise of AutoML is that an automated procedure can find better neural network designs and training schema than manually-designed methods.

Presentation:
* Itemize section 4.1 or de-itemize introduction contributions
* Table 1 is a mess. Notice how "Hyperparameters" and "Search Space" overlap with each other?
* Provide more clarification on the items in Table 1. For instance, are bit options a range? If not, why are the choices for weight bits 2 and 8, yet the default is 5?
* Revise writing for Section 5.1 talking about the A100, as even though it is a very powerful server GPU it is older now.
* Section 5.2.1 => Figures should be introduced in text in the order they appear in the paper.
* Page 8: Generally avoid having whole pages devoted to floats. Additionally, these floats contain a lot of empty white space and can be optimized by shrinking their height, increasing marker size, and making the color of "Uncertainty Bounds" stand out more. The reviewer comments that the font size of these figures is adequate.
* Figure 10 same issue with white space. Potentially consider a logarithmic plot.

**Strengths And Weaknesses:**

Strengths:
* The paper is well-motivated as there is a further need for Green AutoML and applying AutoML techniques to DSNN is useful.
* The related work is appropriately well-written and easy to follow. For the most part, the background and metholodogy are as well.

Weaknesses:
* The proposed methodology is not terribly novel as it is mostly an application of existing off-the-shelf techniques.
* The paper does not really compare against baseline approaches in the literature.
* Aspects of this work such as the Configuration Space arguably lie more in the realm of Neural Architecture Search (NAS) [1] than HPO, yet no discussion of this subfield is conducted.
* The paper primarily conducts experiments on an Nvidia A100 GPU using very old models that severely under-utilize the GPU. For instance, the authors routinely mention their usage of ResNet-20 which is a very small and not even considered in the ResNet Strikes Back [2] paper.
* The reviewer would question some of the provided insights, e.g., Figure 8 hyperparameter importance w.r.t. loss - why is epochs listed there at all? Carbon cost does scale directly with epochs for sure but it is irrelevant to inference once a model is trained.
* The presentation is severely lacking in parts. Will elaborate in requested changes.

---

> ### Author Response · Authors · 2025-04-11
>
> Dear Reviewer, thank you very much for your detailed feedback. We will answer your review point by point.
>
> > The proposed methodology is not terribly novel.
>
> To the best of our knowledge, novelty is not the main criteria for TMLR but technical correctness and the overall importance of the findings.
>
> > The paper does not really compare against baseline approaches in the literature.
>
> We note that the goal of our paper is not to propose a new AutoML approach but to show the potential of AutoML on DSNNs. By following this line of thought, the most relevant baseline is the original configuration of the DSNN authors. We are not aware of any other paper in which AutoML was systematically applied to DSNNs.
>
> > Aspects of this work, such as the Configuration Space, arguably lie more in the realm of Neural Architecture Search (NAS) [1] than HPO.
>
> We agree that roughly half of our configuration space deals with hyperparameters related to the architecture of the DSNN. We point this out in our introduction and cite, to the best of our knowledge, the most cited survey on NAS. Since we do not use any kind of evolutionary algorithm in our paper, we do not understand the reference to the RetNAS paper by Xiao et al. We cite all relevant references and explain all the underlying methods used in our paper.
>
> > The paper primarily conducts experiments on an Nvidia A100 GPU using very old models that severely under-utilize the GPU.
>
> We fully agree with the point, unfortunately we cannot offer any alternative hardware. Nevertheless, from previous work with collaboration partners, we believe that also smaller/older GPUs following the same overall Nvidia architecture patterns will benefit from our results.
>
> > Figure 8 hyperparameter importance w.r.t. loss - why is epochs listed there at all?
>
> Since we use epochs as the fidelity type, our analyzed meta data also includes this a possible factor. As you correctly point out, it is not of direct relevance and thus we removed it from the plot.
>
> > Use larger and more powerful models that fully take advantage of the A100 GPU.
>
> The goal of our DSNN optimization is to find optimal, computationally inexpensive architectures. We aim to facilitate the efficient use of edge devices and enable computations in IoT environments (see Sections 1 and 6 in our paper), thus, use cases with strict limitations on computational availability and energy. In this context, applications such as automated driving functions still heavily rely on lightweight CNNs, incl. smaller ResNets (Fang et al. (2023), Yang et al. (2023)). Therefore, we optimize these types of architectures instead of larger models that currently might be out of focus in the relevant circumstances.
>
> > Using an A100 for ResNet-20 and MobileNetV2 is very much overkill and subtracts from the substance of the paper's findings.
>
> As argued above, we expect our results to transfer to smaller hardware. While the A100 isn't fully utilized, this doesn't affect our approach, as we measure energy consumed during inference. Any constant overhead from idle kernels doesn't negate our absolute energy reductions.
>
> > Experiment with smaller GPUs, potentially NPUs, ASICS, etc.
>
> We fully agree that it would be quite nice to run experiments on more modest hardware including NPUs and ASICS. Unfortunately, these are not available to us. Nevertheless, most importantly, despite reducing the capacity of the hardware, our approach will work analogously. We do not have any indication that this would affect the insights we gained, especially the optimal configurations identified by our AutoML approach to balance performance and efficiency.
>
> > Presentation
>
> We adjusted the itemization and Table 1 accordingly. For clarification: The search space for the weight bits is [2,8], which is commonly a closed interval from the integers 2 to 8, hence 5 being the default. We do not see the need to adjust this in the paper since this is standard mathematical notation.
>
> We noted that the A100 has long been a standard in industry and research (see Section 5.1) and better integrated the figures into the text. We kept the structure of Figure 1, as splitting it would hinder visual interpretability due to whitespace constraints in the TMLR template. We deem marker size and the color of the uncertainty bounds appropriate for the figures.
>
> For Figure 10, we would like to emphasize the outliers with high emission values since this is the main point of the findings. We believe that this is more prominent on non-log scale.
>
> We sincerely hope that these improvements will alleviate your concerns and enable you to accept our work.
>
> Fang, S., Zhang, B., & Hu, J. (2023). Improved mask R-CNN multi-target detection and segmentation for autonomous driving in complex scenes. Sensors, 23(8), 3853.
>
> Yang, L., Lei, W., Zhang, W., & Ye, T. (2023). Dual-flow network with attention for autonomous driving. Frontiers in Neurorobotics, 16, 978225.

---

> > ### Comment · Reviewer_8oNe · 2025-05-22
> >
> > I thank the authors for their response. While I appreciate the efforts made to improve the presentation of the paper and the EfficientNet results provided for reviewer dPd2, these changes made so far are not sufficient to recommend acceptance.
> >
> > In particular, the justification the authors provide for not being able to comply with requested changes, specifically the comments on the transferability and generalizability of their technique to other hardware, as well as the lack of a proper baseline, are not convincing.
> >
> > In general, the biggest issue facing this paper is an experimental setup that is not well designed as it consists of mismatched hardware and models, which give rise to findings that are not very useful nor believably generalizable unless a large suspension of disbelief is employed.

---

> ### Author Response · Authors · 2025-05-23
>
> Dear Reviewer,
>
> Thanks for coming back to us. Unfortunately, we are not sure how to handle your response. Could you please clarify why our comments on transferability and generalizability are not convincing for you? We also appreciate any literature reference showing that results on high-end GPUs (e.g., A100 or H100) do not translate to similar small-scale hardware (e.g., Nvidia Jetson). To dig deeper into it, we talked with colleagues from chip design and they are also not aware of any differences (unless one goes for very special chips that do not follow the common Nvidia design patterns).
>
> (As some background information: we do not have this small-scale hardware readily available to us. Therefore, we are a bit hesitant to go for that without any concrete evidence since this would imply we have to buy further hardware and set it up.)

---

### Review · Reviewer_xxxD · 2025-04-16

**Summary Of Contributions:**

This work presents an investigation into the use of AutoML approaches for hyperparameter optimisation applied to deep shift neural networks (DSNNs) in the pursuit of finding highly energy efficient networks and network parameter regimes. A novel combination of multi-fidelity hyper parameter optimization combined with a multi-objective optimization process attempts to find ideal Pareto tradeoffs for DSNN design.

**Audience:**

Yes

**Broader Impact Concerns:**

None.

**Claims And Evidence:**

No

**Requested Changes:**

For a longer discussion on these, please see the weaknesses section above. To make things more concise:

- To understand the results better, I would request that not only loss but also accuracies are provided
- A more full explanation of how the hyperparameter importance changes across different models would be useful. Also to answer if there is a consistency across models.
- The cost of the search for the Pareto front should also be considered (or at least discussed) so that one may take this into consideration when finding models.
- Aside from the default DSNN models, no alternative approaches to DSNNs are compared against as baselines. Appropriate baselines should be used so that the significance of these contributions can be placed in context.
- The datasets used are extremely small and limited, an analysis of how well these approaches work as we scale up closer to real world datasets would answer whether this methods scales or not. ImageNet scale would be appropriate for understanding scale for a sufficiently difficult task.

**Strengths And Weaknesses:**

__Strengths:__
The intention and potential impact of this work is clearly important in an age where energy usage of AI models in training and inference are major issues for sustainability. The approach also covers a reasonable amount of search space in attempting to find better hyperparameters for these models.

__Weaknesses:__
The paper is rather hard to pin down in terms of how one might understand the outcomes. First, although it is often claimed that high accuracy is achieved, in fact the loss is the only metric described so the accuracy of the models tested are unclear. Furthermore, it is unclear how much the cost of the hyperparameter optimization was in order to achieve the final (performant) configurations or to even find the Pareto front.

Beyond this, the hyperparameter importances seem to suggest that loss and emissions have rather unrelated hyperparameter importances. What to do with this is also unclear. Not to mention that it is unclear whether the same hyperparameters are important across models, therefore again: what is the take-away?

Finally, and perhaps most importantly, it seems that there are no alternative baseline approaches (ourside of DSNNs) that the results are compared against. This means that the take-away, in terms of how well DSNNs can do in place of alternative quantization methods, distillation methods, or otherwise, remains unclear. For me (someone outside of the specific niche), this gives very little understanding of how to place the contribution of this work in general and how to transfer it's lessons outside of this work.

---

> ### Author Response · Authors · 2025-04-26
>
> Dear reviewer,
> Thank you for your insightful feedback. We will answer each comment individually:
>
>
> > To understand the results better, I would request that not only loss but also accuracies are provided
>
> We would like to clarify that, since we use the standard classification loss, loss = 1 - accuracy holds (as stated at the end of subsection 5.1). Since the AutoML problem is often formulated as a minimization problem, we opted for reporting the loss instead of accuracy directly. If you believe that this could be a problem for readers, we are nevertheless happy to change this.
>
> > A more full explanation of how the hyperparameter importance changes across different models would be useful.
>
>
> Thank you very much for this suggestion. We agree that a more detailed analysis of how hyperparameter importance changes across different models is very valuable, particularly to assess consistency. In response, we added additional plots in the appendix showing the hyperparameter importance for each architecture we examined. As discussed in the paper, the DSNN-specific hyperparameters consistently remain among the most important across different models, together with weight decay and learning rate, supporting the relevance of our analysis to present insights into DSNNs.
>
> > The cost of the search for the Pareto front should also be considered.
>
> We fully agree that this is an important discussion. To provide sufficient evidence, we provide an upper bound calculation based on the experiments we ran. Our longest AutoML optimization run—GoogLeNet on Caltech101—took approximately 48 hours to determine an approximated Pareto front, which is by far the longest; most other runs were completed in about half that time. Assuming a maximum 300W power draw on an A100 GPU according to Nvidia and a local carbon intensity factor of 0.475 kgCO₂/kWh, this results in an estimated emission of about 6.84 kgCO₂eq, based on the CodeCarbon formula:
>
> Emissions (kgCO₂eq)
> = (0.3 kW×48 h×0.475kgCO₂/kWh)
> ≈ 6.84 kgCO₂eq.
>
> For GoogLeNet, our optimized model reduces inference emissions by approximately 0.0004 kgCO₂eq per inference—meaning the optimization overhead is amortized after around 17,100 inferences, or about 4.75 hours assuming a conservative real-time inference rate of one image per second (which is much slower than actual rates in automated driving or production contexts, which is what we are targeting). For other models in our study, the emission savings per inference are considerably higher, meaning the break-even point is reached even sooner.
>
> > Aside from the default DSNN models, no alternative approaches to DSNNs are compared against as baselines.
>
> In this work, our primary focus is to analyze and gain deeper insights into the behavior and performance characteristics of DSNNs, as introduced in the original DSNN paper [Elhoushi et al. (2021)]. We do not aim to provide a broad comparison across quantization or compression methods and we also make no statement regarding state-of-the-art performance, but rather to study this specific DSNN architecture in depth. We believe that it is a very valuable insight for the community that DSNN could perform much better than previously reported (also by the original authors).
>
>
> > The datasets used are extremely small and limited, an analysis of how well these approaches work as we scale up closer to real world datasets would answer whether this methods scales or not.
>
> In view of the above discussion on when AutoML tuning is amortized, it is important to discuss how our results would be transferable to more difficult tasks. We note that we already used efficient approximation techniques, such as multi-fidelity techniques, to limit the AutoML tuning efforts as much as possible. Therefore, we deem it most interesting to study whether our results on the smaller CIFAR10 can be directly translated to the larger ImageNet. (We note that this is an established kind of study in the field of neural architecture search; for example, also done in the inaugural paper on DARTS: Differentiable Architecture Search by Liu ICLR’19).
> Based on our current findings on the CIFAR-10 ResNet20 Pareto front, we expect our results to remain consistent at larger scales and have started experiments to demonstrate this on ImageNet data. Due to computational constraints, completing this evaluation will take some additional time, but we anticipate having results within the next two weeks. Nevertheless, we can already report that several of the configurations on the Pareto front in CIFAR-10 perform better or equal to the default DSNN configuration, but our configurations save 10% emissions. We appreciate your understanding and look forward to sharing the full results.
>
> Elhoushi, M., Chen, Z., Shafiq, F., Tian, Y. H., & Li, J. Y. (2021). Deepshift: Towards multiplication-less neural networks. In Proceedings of the IEEE/CVF conference on computer vision and pattern recognition (pp. 2359-2368).

---

> > ### Author Response · Authors · 2025-05-15
> >
> > Dear reviewer,
> >
> > We appreciate your patience. Based on your feedback, we have now included an additional analysis that directly addresses the question of transferability of our results on other datasets, specifically, Imagenet, and whether configurations found to be optimal on one dataset retain their effectiveness in another.
> >
> > To evaluate this, we took the Pareto-optimal configurations identified for training a ResNet20 on CIFAR-10 and applied them to train a ResNet on ImageNet100, a standard subset of ImageNet. The results are as expected: the default configuration is not part of the new Pareto front and is dominated by nearly all transferred configurations in one or both objectives (loss and emissions). While not all original Pareto-optimal configurations remain Pareto-optimal on ImageNet100, many are very close and still significantly outperform the default.
> >
> > This indicates that the configurations found in one dataset setting can generalize effectively to another, offering strong performance and efficiency without the need for a full re-optimization. This has meaningful implications for low-resource and sustainable AutoML practice, where reusing high-quality configurations can save considerable time and energy.
> > We have added these findings to the paper in Section 5.3.

---

> > > ### Comment · Reviewer_xxxD · 2025-05-21
> > >
> > > Thank you for your detailed response. A number of my concerns are alleviated. However, the argument given against the inclusion of baselines seems unreasonable to me. To understand the impact of this work, and to appreciate whether the improvements measured for DSNNs are important to consider, some context is crucial (note that even the original Elhoushi et al (2021) paper gives comparisons against other methods). It is not necessary for the baselines to have been run from scratch, these could instead be simply taken from existing work.

---

> > > > ### Author Response · Authors · 2025-05-23
> > > >
> > > > Dear reviewer,
> > > >
> > > > We are happy to hear that we could alleviate most of your concerns. We also appreciate your perspective and will thus include the requested baseline provided in the original paper by Elhoushi et al. This includes a VGG19-Small model compared to AdderNet, for which we will run additional optimization runs to match the baseline. We will provide the additional results at the end of next week at the latest. Thank you for your patience.

---

> > > > > ### Author Response · Authors · 2025-05-30
> > > > >
> > > > > Dear reviewer,
> > > > >
> > > > > in response to your suggestion, we have explicitly addressed additional baselines from literature in the revised version of our paper (see Appendix A.3, especially Tables 2 and 3).
> > > > >
> > > > > We compare our method against the same baselines used in the original DSNN paper (cp. Table 2 in the paper by Elhoushi et al.). It includes an unquantized VGG19, the default DeepShift-PS baseline for VGG19, and two external quantized models (AdderNet and ShiftAddNet). Since our method jointly optimizes for accuracy and emissions, but the referenced baselines are only available in terms of accuracy, we restricted the shown evaluation to the accuracy objective for a fair and consistent comparison.
> > > > >
> > > > > We focus on the VGG19 architecture, as this is the architecture for which the original DSNN authors provide benchmark results. From our MOMF results, we selected a Pareto-optimal configuration that improves upon the default configuration. As shown in Table 2, this configuration outperforms all baseline models, including the unquantized original VGG19, indicating that quantization—when properly optimized—can yield superior performance even under constrained emissions.
> > > > >
> > > > > Furthermore, we emphasize that the optimal configuration is only partially quantized (11 out of 19 convolutional layers), which highlights the importance of automated methods like ours. This insight is consistent with our broader analysis in Chapter 5, showing that full quantization is not necessarily optimal. These findings underline the non-trivial nature of designing efficient DeepShift configurations and validate the utility of our method in discovering non-obvious yet effective architectures.
> > > > >
> > > > > We believe this comparison and analysis make a case for the relevance of our approach. We sincerely hope that we have addressed your concern by demonstrating both competitive performance and insight.

---

### Review · Reviewer_dPd2 · 2025-05-01

**Summary Of Contributions:**

In this paper, the authors take Deep Shift Neural Networks (DSNN), a class of deep models, and empirically analyze their competitiveness in terms of a trade-off between performance (accuracy) and energy consumption. Evidently, this requires hyper-parameters research, which is conducted through known multi-fidelity hyper-parameter optimization tools, adapted to handle two objectives. The authors benchmark ResNet20, MobileNetv2 and GoogLeNet on CIFAR-10 and Caltech101, finding in certain cases unexpected trade-offs.

**Audience:**

Yes

**Claims And Evidence:**

No

**Requested Changes:**

Referring to the weaknesses section, please highlight better the technical contribution in case it is not simply limited to Sec. 4.2, and upgrade the experimental section following the indicated guidelines in the weaknesses.

Are you going to release the code open-source in case of acceptance of the work?

**Strengths And Weaknesses:**

## Strengths
- Overall, Deep shift neural networks are not as explored as other approaches, and this work's aim is to advance our knowledge in these architectures.
- DSNNs are promising to reduce computation, and the overall motivation behind this work is valid.
- The work is pointing out specific efficient configurations in the tested designs (this is also a weakness).
- The work does a great job of reviewing the sota and providing the background.

## Weaknesses
- The overall technical contribution is weak. This work indicates specific configurations found in specific setups (a strength), but this is very limiting when claiming generality, given that the finding is limited to the specific setups presented in the work. The only possible technical novelty lies in (4.1), but I am wondering how that is not straightforward.
- The experiments are very limited, the architectures are outdated, and the datasets are small-scale. I doubt it is possible to claim any generality from the observations taken from these architectures/datasets. Especially in a context of energy consumption analysis, more recent architectures (like MobileNetv3) or more effective (like the EfficientNet family, or NAS-generated ones) should be generated. Also, Transformers are becoming more and more present - since the approach is not limited to convolutional architectures, these should be benchmarked too.
- The only analyzed task is image classification - is there any reason for that ? Showcasing applicability on efficient architectures deployed on different tasks, like YoLOv11 on object detection, would showcase the broad applicability.
- There is a possible background pitfall in the whole work: this work is not really minimizing the CO2 emissions, but rather the *energy consumption*. Indeed, as the same authors write in Sec. 5.1, "we use the CodeCarbon emissions tracker to track carbon emissions from
computational processes, in our case inference, by monitoring energy use and regional energy mix in kgCo2eq"; hence, the estimated energy consumption is multiplied by a factor to get the CO2 emitted in that specific region. If the algorithm minimizes CO2, it would also take into account *how* the energy is produced, *how far* the machine is from the production site, etc., which is not really what is done in this work. *This is easily fixable, just claiming that the algorithm aims to reduce power consumption, which also translates into CO2 emissions reduction*.


### Minor Weaknesses
- *Readability*. The authors should carefully check that all the definitions and abbreviations are correctly provided - for example, in the abstract, HPO is not defined (yes it is hyperparameter optimization, but it should be defined)

---

> ### Author Response · Authors · 2025-05-12
>
> Dear reviewer,
>
> Thank you for your suggestions. In general, we have the impression that your biggest concern is on the generalizability to other models, datasets, or tasks. First of all, we like to highlight that our focus is on low-energy applications and thus, we deem large-scale transformer models not relevant for our case. Furthermore, it is very close to our hearts to limit the experiments as much as possible to not unnecessarily waste compute resources and CO2eq emissions without need. We fully see the point that it could be interesting to benchmark an approach on yet another dataset, a larger model, or other confounding factors. However, from the extensive experiments on several models and two datasets, we haven’t seen any evidence that our results do not generalize. We are happy to run more experiments to validate our results, if there are concrete indicators from a mathematical point of view or other empirical studies that would suggest that our results might not always hold. We kindly ask you to provide these in such a case.
>
>
> > This work indicates specific configurations found in specific setups (a strength), but this is very limiting when claiming generality, given that the finding is limited to the specific setups presented in the work.
>
> We demonstrate our approach across multiple architectures and two datasets, explicitly to highlight its general applicability beyond a single setup. Moreover, the approach itself is designed to be architecture- and task-agnostic, and we clarified in the paper that it can be applied to other models or data with no methodological changes (Section 4.2).
>
> As for the experimental setup: While we use an A100 GPU for experimentation, our approach measures power consumed during inference. Hence, we strongly expect our method - and the resulting energy savings - to transfer to other hardware without fundamental change.
>
> > Especially in a context of energy consumption analysis, more recent architectures (like MobileNetv3) or more effective (like the EfficientNet family, or NAS-generated ones) should be generated.
>
> Our work primarily targets latency- and energy-constrained environments, such as those encountered in autonomous driving. In these settings, practical applications heavily rely on lightweight convolutional networks, including compact ResNet variants (e.g., Fang et al., 2023; Yang et al., 2023). For this reason, our study focuses on optimizing such architectures.
>
> We appreciate the reviewer’s suggestion and will extend our experiments to include EfficientNetV2; we will provide the corresponding results as soon as possible.
>
> > The only analyzed task is image classification - is there any reason for that ?
>
> We agree that demonstrating applicability to additional tasks like object detection could further highlight the generality of our approach. However, we intentionally focused on image classification for several reasons:
>
> - Relevance: Image classification remains a core task in computer vision. It enables reproducible comparisons across methods with minimal confounding task-specific factors.
> - Backbone Transferability: Many object detection models (including YOLO variants and Faster RCNN) build on classification backbones such as ResNet or MobileNet. Optimizations in classification translate directly to improvements in perception pipelines, particularly in terms of speed and energy efficiency.
>
> > There is a possible background pitfall in the whole work: this work is not really minimizing the CO2 emissions, but rather the energy consumption.
>
> We appreciate the reviewer’s observation regarding the distinction between minimizing energy consumption and directly minimizing carbon emissions. Our method targets energy-efficient model configurations and uses CodeCarbon to estimate resulting emissions based on energy consumption and the regional energy mix. While there is no way to control for external factors such as energy production methods or transmission losses, our goal aligns with widely accepted practices in the ML sustainability community, where energy minimization is a practical and measurable proxy for reducing carbon emissions [Strubell et al., 2019; Henderson et al., 2020].
> That said, we revised our paper to make this distinction clearer (Section 5.1). We claim that our algorithm minimizes energy consumption, which, when paired with emission models, correlates strongly with CO2 reduction, especially in real-world deployment scenarios.
>
> > Are you going to release the code open-source?
>
> Yes, we promise that the code will be released open-source.
>
> References:
> Strubell at al. (2020). Energy and policy considerations for modern deep learning research.
>
> Henderson et al. (2020). Towards the systematic reporting of the energy and carbon footprints of machine learning.
>
> Fang et al. (2023). Improved mask R-CNN multi-target detection and segmentation for autonomous driving in complex scenes.
>
> Yang et al. (2023). Dual-flow network with attention for autonomous driving.

---

> > ### Author Response · Authors · 2025-05-15
> >
> > Dear Reviewer,
> >
> > thank you very much for your patience.
> >
> > We are now happy to include the results of our EfficientNetV2 analysis in Section 5.2.1 of the revised manuscript. We agree that evaluating EfficientNetV2 is relevant, as it represents a current architecture explicitly designed for performance and efficiency.
> >
> > To that end, we applied our MOMF approach to EfficientNetV2 on the CIFAR-10 dataset, using multiple random seeds to ensure robustness. As expected, we were able to identify new Pareto-optimal configurations that improve upon both prediction loss and inference-time emissions, in line with the trends observed in our previous experiments. Notably, while EfficientNetV2 already incorporates efficiency-focused design principles, our results highlight the continued importance of carefully tuning DSNN-related hyperparameters. Compared to the default configuration, we achieved more than a 20% reduction in emissions at inference without sacrificing accuracy.
> >
> > These findings further support the generalizability and effectiveness of our approach in optimizing resource efficiency across deep learning architectures.

---

> > ### Comment · Reviewer_dPd2 · 2025-05-22
> > **Thank you for the rebuttal**
> >
> > I would like to thank the authors for their answers; and importantly, they are back aligning with the fact that they are actually minimizing power consumption, which is important!
> >
> > I have gone through the reviews and answers provided, and I appreciate the author's effort to include EfficientNet. However, i still feel the argument the authors are providing is insufficient to claim generality while the only task explored is image classification. Furthermore, there are some works that are (trying to) deploy Transformer-based architectures in resource-constrained scenarios [A-D] and beyond - as such, the argument provided by the authors to this regard seems not valid.
> >
> > I will be happy to engage in further discussion in case the authors have counterarguments that differ from what has already been provided in their rebuttal.
> >
> > [A] Lin, Ji, et al. "Awq: Activation-aware weight quantization for on-device llm compression and acceleration." Proceedings of Machine Learning and Systems 6 (2024): 87-100.
> >
> > [B] Nguyen, Le-Trung, et al. "Activation Map Compression through Tensor Decomposition for Deep Learning." The Thirty-eighth Annual Conference on Neural Information Processing Systems.
> >
> > [C] Sarkar, Souvika, et al. "Processing Natural Language on Embedded Devices: How Well Do Transformer Models Perform?." Proceedings of the 15th ACM/SPEC International Conference on Performance Engineering. 2024.
> >
> > [D] Scherer, Moritz, et al. "Work In Progress: Linear Transformers for TinyML." 2024 Design, Automation & Test in Europe Conference & Exhibition (DATE). IEEE, 2024.

---

> ### Author Response · Authors · 2025-05-23
>
> Dear reviewer,
>
> Thank you for your continual feedback. Your last comment helped clarify the generality issue for us. We are sorry for the misunderstanding – we did not intend to claim generality across all computer vision tasks. In our view, the experiments are sufficient to support generalizable insights for DSNNs within the scope of image classification. We updated the paper to now clearly state in the abstract, introduction, and limitations that our focus is solely on image classification (blue text). We believe this scope is justified, as image classification is a crucial component in many detection pipelines. We appreciate your feedback in helping us make this clearer in the paper.
>
> We also appreciate the reviewer’s effort in providing relevant references pertaining to transformer architectures and were happy to examine them in detail. While the works offer valuable insights into the role of (quantization of) transformers in automated driving, we found that two of them [A, C], primarily focus on language processing, which lies outside the core scope of our expertise and current project. That said, we fully acknowledge that quantization and efficient deployment of transformers represent a highly promising direction—one that could indeed merit a dedicated investigation in another project. (Note: Reference [B] is only doing experiments on CNNs, as we do, and only mentions vision transformers as a side note.)
>
> Given our focus on image classification on edge devices, we deliberately scoped our study around CNNs, which still remain highly relevant in edge-based image classification scenarios. To ensure clarity, we now explicitly state this in the limitations section, both to define the scope of our work and to recognize the importance of this evolving area (blue text).
> (We would like to point out that compared to [D] with improvements between 0.1 and 0.9 accuracy points, we achieve improvements on the best known DSNN settings between 5 and 25 accuracy points (Figure 1).)